# Decompose, Structure, and Repair: A Neuro-Symbolic Framework for Autoformalization via Operator Trees

Xiaoyang Liu[1]  Zineng Dong[1]  Yifan Bai[1]  Yantao Li[2]  Yuntian Liu[1]  Tao Luo[1][3]

## Abstract

Statement autoformalization acts as a critical bridge between human mathematics and formal mathematics by translating natural language problems into formal language. While prior works have focused on data synthesis and diverse training paradigms to optimize end-to-end Large Language Models (LLMs), they typically treat formal code as flat sequences, neglecting the hierarchical logic inherent in mathematical statements. In this work, we introduce *Decompose, Structure, and Repair (DSR)*, a neuro-symbolic framework that restructures autoformalization into a modular pipeline. DSR decomposes statements into logical components and maps them to structured operator trees, leveraging this topological blueprint to precisely localize and repair errors via sub-tree refinement. Furthermore, we introduce PRIME, a benchmark of 156 undergraduate and graduate-level theorems selected from canonical textbooks and expertly annotated in Lean 4. Experimental results demonstrate that DSR establishes a new state-of-the-art, consistently outperforming baselines under equivalent computational budgets. The datasets, model, and code are available at https://github.com/XiaoyangLiu-sjtu/DSR.

## 1. Introduction

Formal mathematics leverages interactive theorem provers (ITPs) such as Isabelle (Paulson, 1994), HOL Light (Harrison, 1996), Rocq (Barras et al., 1997), and Lean (de Moura et al., 2015) to provide foundational rigor and absolute correctness for mathematical reasoning. Despite their potential, the widespread adoption of ITPs is hindered by the steep learning curve of formal languages and the labor-intensive manual formalization. To bridge this gap, statement autoformalization has emerged as a promising field, aiming to automatically translate statements from natural language into their formal counterparts (Szegedy, 2020).

While recent advancements have evolved from simple neural machine translation to sophisticated LLM-driven paradigms, most existing approaches fundamentally treat formalization as a monolithic, end-to-end sequence generation task. By mapping natural language (NL) directly to formal language (FL) as flat strings, these models often neglect the hierarchical structure inherent in mathematical statements. This underscores the potential of incorporating explicit intermediate representations into the autoformalization process, which enhance both the semantic fidelity of generation and the precision of error localization.

To address this, we propose *Decompose, Structure, and Repair (DSR)*, a neuro-symbolic framework that decouples the formalization process into distinct, modular stages. Instead of a direct translation, DSR first decomposes the input **NL statement** into **NL components**, comprising distinct conditions and conclusions. These components are then translated into linear code fragments, termed **FL components**, and simultaneously into **FL OPTs**, which are operator trees (OPTs) representing the logical structure of the formal code. Crucially, this structural alignment deepens the model's autoformalization capability, while providing a topological blueprint for our tree-guided repair strategy to precisely localize and correct errors.

We further introduce *Problem Repository of Instructional Mathematics for Evaluation (PRIME)*, a benchmark designed to evaluate autoformalization across a diverse spectrum of mathematical domains, spanning undergraduate to graduate levels. Encompassing fields such as Algebra, Analysis, and Number Theory, PRIME comprises 156 theorems manually formalized by Lean experts from canonical textbooks. These expert annotations ensure strict semantic consistency between NL statements and FL statements. Consequently, PRIME serves as a rigorous and highly reliable gold standard for assessing formalization capabilities.

[1]School of Mathematical Sciences, Shanghai Jiao Tong University [2]Zhiyuan College, Shanghai Jiao Tong University [3]Institute of Natural Sciences, MOE-LSC, CMA-Shanghai, Shanghai Jiao Tong University. Correspondence to: Tao Luo <luotao41@sjtu.edu.cn>.

*Proceedings of the 43rd International Conference on Machine Learning*, Seoul, South Korea. PMLR 306, 2026. Copyright 2026 by the author(s).

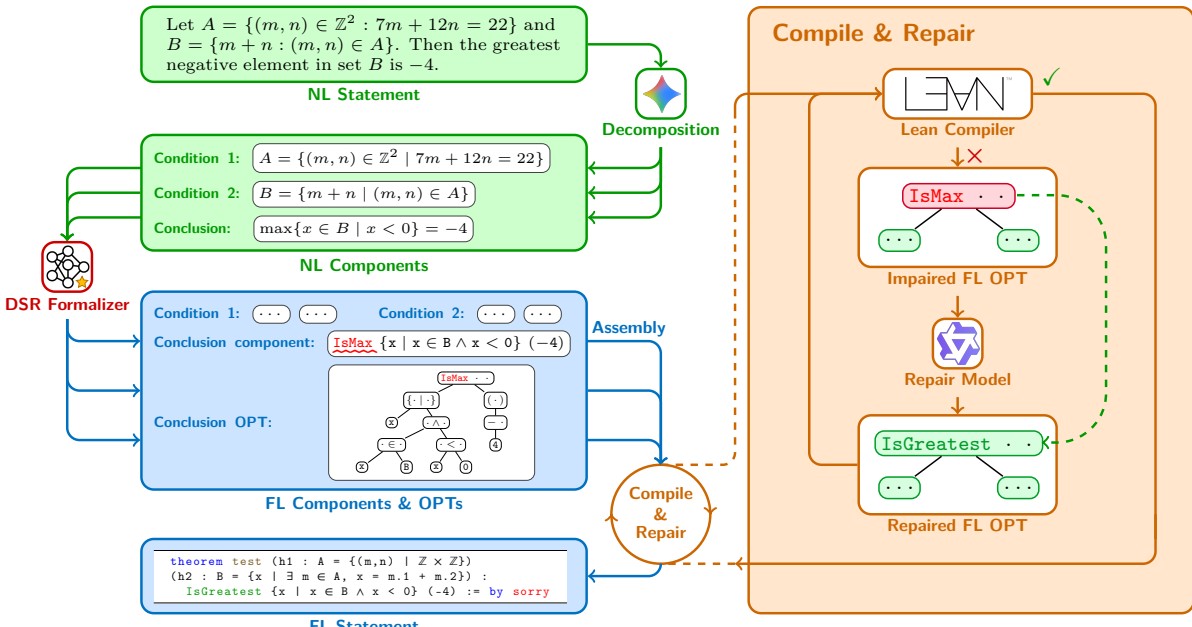

*Figure 1.* **The Decompose, Structure, and Repair (DSR) Framework.** Given an NL statement, DSR first decomposes it into logical NL components. The framework then structures the translation by mapping each NL component to its corresponding FL component and its associated FL OPT. Finally, a tree-guided repair loop leverages the OPT to precisely localize and repair errors via sub-tree refinement.

Extensive evaluations across ProverBench, ProofNet, and the proposed PRIME benchmark rigorously confirm that DSR establishes a new state-of-the-art in autoformalization, consistently achieving the highest Syntax Check (SC) and Consistency Check (CC) pass rates. Crucially, ablation studies validate the efficacy of our training strategy: incorporating operator tree supervision provides essential structural priors, while the integrated curriculum learning effectively guides the model to master these complex hierarchical outputs. Finally, compared to traditional statement-level repair, our tree-guided repair strategy effectively corrects local errors without disrupting the global logic.

Our main contributions are as follows:

1. We propose DSR, a novel neuro-symbolic framework that leverages operator trees to explicitly capture logical structures, significantly enhancing autoformalization and enabling precise, tree-guided repair.
2. We introduce PRIME, a rigorous benchmark comprising 156 expert-formalized theorems from advanced mathematical domains, serving as a highly reliable gold standard for evaluating formalization capabilities.
3. Extensive evaluations demonstrate that DSR establishes a new state-of-the-art across multiple challenging datasets, consistently outperforming significantly larger models under equivalent computational budgets.

## 2. Related Work

**Autoformalization.** Statement autoformalization has evolved from neural machine translation (Wang et al., 2018; Cunningham et al., 2022) to LLM-driven paradigms, progressing from few-shot prompting (Wu et al., 2022; Agrawal et al., 2022; Zhou et al., 2024) to supervised fine-tuning on formal libraries (Gao et al., 2025; Lu et al., 2024; Liu et al., 2025a; Wu et al., 2025; Yu et al., 2025b). To further refine generation quality, contemporary frameworks have integrated reinforcement learning (Yu et al., 2025a; Huang et al., 2025), retrieval-augmented generation (Zhang et al., 2024; Lu et al., 2025), and tool-integrated feedback (Guo et al., 2025) for enhanced correctness. Recently, the field has shifted from one-pass generation to system-level iterative architectures, exemplified by ARIA's graph-of-thought planning (Wang et al., 2025b) and SITA's structure-to-instance instantiation (Li et al., 2025).

While both DRIFT (Zhang et al., 2026) and DSR recognize decomposition as a key driver for advancing autoformalization, they tackle orthogonal challenges. Specifically, DRIFT's decomposition aims to optimize external concept retrieval, whereas DSR's decomposition serves to mathematically reduce the dimensionality of the proposition, thereby enabling the subsequent operator tree generation and tree-guided repair stages. Integrating DRIFT's retrieval mechanism with DSR's structural repair represents a highly promising direction for future autoformalization agents.

**Operator Trees.** Operator Trees (OPTs) model mathematical expressions as hierarchical structures, with operators as internal nodes and operands as leaves (Zanibbi et al., 2002). Departing from flat, sequence-based representations, OPTs explicitly encode operator precedence and logical scoping, a property that underpinned Mathematical Information Retrieval (MIR) systems for structural similarity search (Zanibbi & Blostein, 2012; Hu et al., 2013; Zhong & Zanibbi, 2019) and hybrid indexing (Kristianto et al., 2016). In deep learning, OPTs have been integrated into pretraining to enhance semantic understanding (Peng et al., 2021) or used as direct inputs for formula encoders (Wang et al., 2021). Recently, this paradigm has been adapted to the domain of formal mathematics, where OPT-based metrics are employed to evaluate the semantic equivalence of formal statements (Liu et al., 2026; 2025b).

# 3. Methodology

In this section, we introduce *Decompose, Structure, and Repair (DSR)*, a neuro-symbolic framework for autoformalization as illustrated in Figure 1. The pipeline proceeds in three stages. First, Section 3.1 outlines the decomposition of NL statements into NL components. Next, Section 3.2 details the translation of these components into FL components and FL OPTs. Finally, Section 3.3 presents the tree-guided repair strategy to ensure the syntactic validity and logical correctness of the generated Lean code.

## 3.1. Decomposing Statements into Components

NL statements are typically optimized for human readability, relying on implicit context and flexible phrasing that might hinder autoformalization. To bridge the gap between this linguistic informality and the rigor required by proof assistants, we introduce a semantic decomposition stage consisting of *Semantic Canonicalization* and *Structural Role Alignment*. Our objective is to preprocess the raw NL statement into a sequence of refined NL components, thereby simplifying downstream formalization tasks. Figure 2 demonstrates this process using an example from ProofNet (`exercise_26_12`).

**Semantic Canonicalization.** The first task, *Semantic Canonicalization*, focuses on resolving the ambiguity and redundancy inherent in natural language to produce a Lean-friendly input for formalization. Specifically, this process involves two key operations: (1) **filtering linguistic noise** by removing supplementary text that is logically redundant in a formal setting, thereby isolating the core mathematical constraints from rhetorical artifacts; and (2) **explicating implicit context** by recovering omitted variable types or background assumptions that are inferred implicitly by humans but are essential for formal completeness. This stage

yields a canonicalized text representation that serves as a clean, Lean-aligned input for the subsequent alignment step.

**Structural Alignment.** The second task, *Structural Role Alignment*, imposes logical structure upon the canonicalized text. Specifically, this process segments the input into discrete components and classifies each as either a CONDITION or a CONCLUSION. This categorization aligns the definition of NL components with the standard logical structure of mathematical propositions. Crucially, this distinction serves as a structural prior for the subsequent translation stage (Section 3.2). By decoupling CONDITIONS from the CONCLUSION, we enforce a clear syntactic separation. This guides the translation to correctly instantiate conditions as variables or hypothesis binders and the conclusion as the proof goal, thereby preventing logical inversions in the generated Lean code.

While we define canonicalization and alignment as distinct logical stages, in practice, we execute them in a single pass via Gemini 3.0 Pro. The prompt is provided in Appendix D.

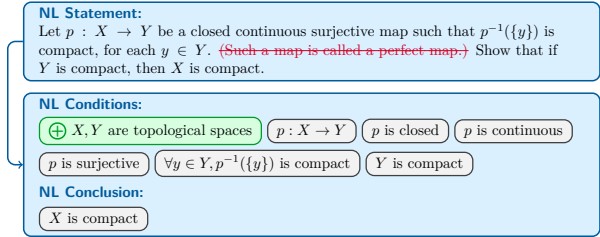

*Figure 2.* **Semantic Decomposition.** The NL statement is decomposed into NL components via *Semantic Canonicalization* to filter crossed-out noise and explicate green-highlighted implicit context, followed by *Structural Role Alignment* to categorize these components into CONDITIONS and CONCLUSION.

## 3.2. Structuring Translation with Operator Trees

### 3.2.1. MOTIVATION

In our framework, the formalizer is designed to generate the FL component and its FL OPT as a joint output sequence. By coupling the formal code with its structural representation, this design offers two distinct advantages.

**Theoretical: Semantic Anchoring.** The core innovation of introducing FL OPTs lies in providing a hierarchical semantic anchor that transcends the limitations of linear sequence generation. Traditional autoformalization treats Lean code as a flat string, often struggling to capture the nested operator precedence and logical dependencies inherent in mathematical expressions. By mandating the joint prediction of an FL OPT, we impose a structural constraint that forces the formalizer to explicitly model the recursive topology of the target FL component. This ensures the

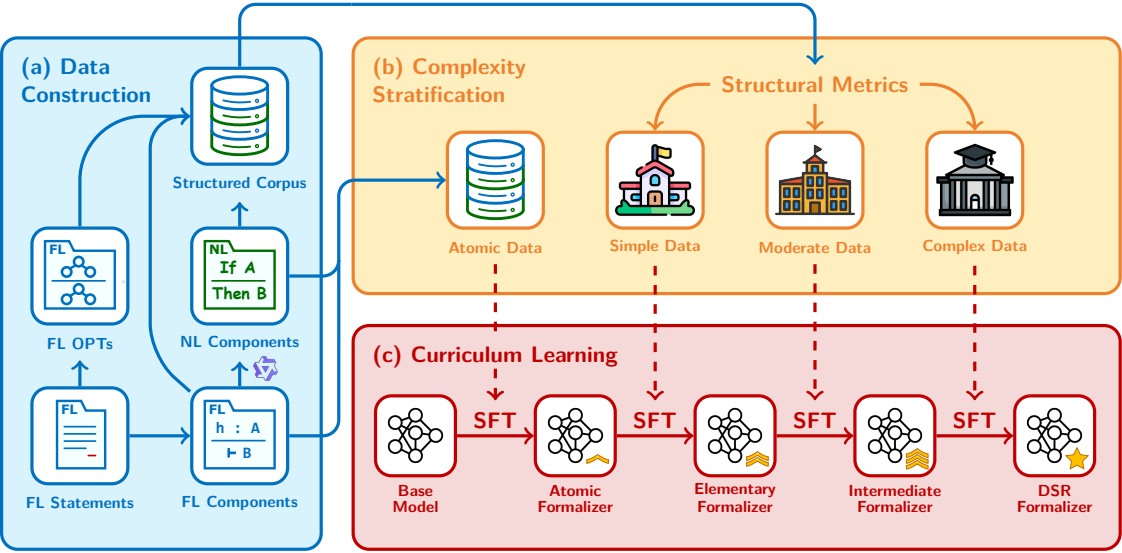

*Figure 3.* **Training Pipeline of the DSR Formalizer.** (a) Data Construction: A structured corpus is built by aligning NL components, FL components, and FL OPTs. (b) Complexity Stratification: The corpus is stratified into three complexity levels, while basic NL and FL components constitute the atomic data. (c) Curriculum Learning: A four-stage strategy where the model first learns basic NL-to-FL component translation, then progresses to the joint generation of FL components and FL OPTs.

model internalizes the component's logical skeleton rather than relying on mere statistical correlations.

**Practical: Repair Blueprint.** Beyond theoretical benefits, the FL OPT functions as a topological blueprint of the FL component, partitioning linear code into addressable logical sub-structures. This modularity serves as the essential prerequisite for our repair stage (Section 3.3). By exposing the underlying logical hierarchy, the OPT facilitates surgical interventions that pinpoint and correct localized errors within specific subtrees. This fine-grained control eliminates the computational redundancy of statement-level repair and prevents unintended modification of the correctly formalized subcomponents.

We empirically substantiate both the theoretical anchoring and the practical blueprint via comprehensive ablation studies in Section 4.4.

### 3.2.2. OPT REPRESENTATION

We adopt the FL OPT representation from ASSESS (Liu et al., 2026) to encode the hierarchical logic, as illustrated in Figure 1. Fundamentally, an OPT is a labeled, ordered tree constructed by querying the Lean Language Server for the nested scopes of elements: operators function as parent nodes, while their arguments serve as ordered children. We refer readers to ASSESS (Liu et al., 2026) for the detailed implementation regarding tree construction.

To align with the DSR framework, we introduce two specific

adaptations. First, we construct FL OPTs at the granularity of FL components rather than FL statements, mirroring our decomposition strategy. Second, we explicitly retain parentheses within the FL OPT structure. Unlike ASSESS, which discards parentheses as redundant structural artifacts, we preserve them to enforce strict token-level consistency between the linear code and the hierarchical tree.

### 3.2.3. DSR FORMALIZER

The training pipeline of the DSR Formalizer, as illustrated in Figure 3, unfolds in three stages: (1) *Data Construction*, (2) *Complexity Stratification*, and (3) *Curriculum Learning*.

**Data Construction.** We construct a component-level parallel corpus sourced from NuminaMath-LEAN (Wang et al., 2025a) and ATLAS-Synthetic (Liu et al., 2025a), totaling 120,627 FL statements. Using the tree construction toolkit provided by ASSESS (Liu et al., 2026), we parse these statements into FL components and generate their corresponding FL OPTs. We then employ Qwen3-Max (Yang et al., 2025) to back-translate these fragments into NL components, using the prompt detailed in Appendix D. The final corpus comprises 283,958 aligned triples of NL components, FL components, and FL OPTs.

**Complexity Stratification.** To enable progressive curriculum learning, we stratify the structured corpus via three structural metrics of FL OPTs: *tree depth*, *tree width*, and *the total number of nodes*. Based on these metrics, we rank

the corpus and filter out the top 1% of samples with extreme complexity. The remaining valid dataset of 281,209 samples is then partitioned into three distinct difficulty tiers, comprising 143,325 simple, 109,829 moderate, and 28,055 complex samples. This complexity stratification ensures that the model is exposed to a smooth gradient of structural difficulty during the training process.

**Curriculum Learning.** We implement the curriculum learning (Bengio et al., 2009) strategy by fine-tuning Qwen2.5-7B-Instruct (Qwen et al., 2025) via LoRA (Hu et al., 2022) over four progressive phases to obtain the DSR Formalizer. As detailed in Table 1, the curriculum transitions from linear code generation (Phase 1) to structural tree prediction of increasing complexity (Phases 2–4). To prevent catastrophic forgetting, we employ a replay mechanism that mixes the full dataset of the current complexity tier with sampled data from previous tiers, thereby balancing new structural knowledge with established syntactic patterns.

*Table 1.* **Hyperparameters and Data Mixing Ratios for Curriculum Learning.** We adopt a replay mechanism where the training batch is composed of the current complexity tier mixed with sampled data from previous tiers.

| Config / Phase | Phase 1 | Phase 2 | Phase 3 | Phase 4 |
|---|---|---|---|---|
| **Focus** | FL components | FL components & FL OPTs | | |
| **Primary Data** | Atomic (100%) | Simple (90%) | Moderate (70%) | Complex (50%) Moderate (30%) |
| **Replay Data** | - | Atomic (10%) | Simple (30%) | + Simple (20%) |
| **Epochs** | 1 | 1 | 1 | 1 |
| **Total Batch Size** | 128 | 128 | 128 | 64 |
| **Learning Rate** | 2e-4 | 1e-4 | 5e-5 | 1e-5 |
| **Warmup Ratio** | 0.03 | 0.10 | 0.03 | 0.03 |

### 3.3. Repairing Errors via Tree-Guided Strategies

The final stage of DSR transforms the predicted FL components and FL OPTs into a verified FL statement. This process comprises two phases: *Structure-First Assembly* and *Tree-Guided Repair*.

**Structure-First Assembly.** Upon generating the output sequence, we reconstruct the FL statement via a deterministic assembly process. Adopting a structure-first strategy, we prioritize the FL OPT by recursively concatenating its leaf nodes to form the final code. We rely on the OPT representation because its hierarchical constraints effectively prevent syntax errors common in linear generation, such as mismatched parentheses or unclosed scopes.

The FL components, while secondary during inference, serve two critical roles: (1) Training Stabilizer: Joint training with linear Lean code provides intermediate supervision that accelerates convergence. (2) Inference Failsafe: In rare cases where the generated OPT is structurally invalid (e.g., a node count mismatch), we discard the malformed OPT and fall back to the FL component to ensure a valid output

string is always produced.

**Tree-Guided Repair.** The assembled FL statement is submitted to the Lean compiler for verification. Upon failure, we leverage the FL OPT to localize errors and execute a hierarchical repair strategy. Instead of repairing the entire statement, we map the compiler's error message (i.e., row and column indices) to the specific tree node corresponding to the failure. We then attempt to resolve the error by sequentially escalating through three levels of granularity.

---

**REPAIR EXAMPLE: ProverBench (`aime_2025ii_p4`)**

**NL Statement:** The product $\prod_{k=4}^{63} \frac{\log_k(5^{k^2-1})}{\log_{k+1}(5^{k^2-4})} = \frac{\log_4(5^{15})}{\log_5(5^{12})} \cdot \frac{\log_5(5^{24})}{\log_6(5^{21})} \cdot \frac{\log_6(5^{35})}{\log_7(5^{32})} \cdots \frac{\log_{63}(5^{3968})}{\log_{64}(5^{3965})}$ is equal to $\frac{m}{n}$, where $m$ and $n$ are relatively prime positive integers. Find $m + n$. Show that it is 106.

**Initial Generation State**

*NL Component:* $\frac{m}{n} = \prod_{k=4}^{63} \frac{\log_k\left(5^{k^2-1}\right)}{\log_{k+1}\left(5^{k^2-4}\right)}$

*FL Component:* `h2 : m / n = ∏ k ∈ Finset.Icc 4` `63, logb k (5^(k^2 - 1)) ...`

**1. Subcomponent-Level Repair**
`logb k (5^(k^2 - 1))`
× *Error: Function 'logb' not found in scope.*
↪ `Real.logb k (5^(k^2 - 1))`
✓ *Fix: Corrected to namespace 'Real.logb'.*

**2. Subcomponent-Level Repair**
`∈ Finset.Icc 4 63`
× *Error: 'Finset.Icc' requires discrete type. $\mathbb{R}$ is not locally finite.*
↪ `∈ Finset.Icc (4:ℕ) 63`
✓ *Fix: Type annotation '(: ℕ)' enforces discrete domain.*

**3. Subcomponent/Component-Level Repair**
✓ *No compilation errors found. Skipped*

**4. Statement-Level Repair**
✓ *No compilation errors found. Consistency check passed.*

---

**Final Verified Code (Snippet):**
`theorem test ... (h2: m / n = ∏ k` `∈ Finset.Icc (4:ℕ) 63, Real.logb k` `(5^(k^2 - 1)) ... := by sorry`

*Figure 4.* **A repair trajectory of the tree-guided repair process.** More detailed repair examples can be found in Appendix B.7.

- **Subcomponent-Level Repair.** We initiate an *iterative bottom-up repair* trajectory. Starting from the immediate parent of the erroneous node, we extract the minimal subtree rooted at the current ancestor for surgical repair. If verification fails, we recursively expand the repair scope to the grandparent. This cycle continues until the repair succeeds or the scope reaches the component boundary.

- **Component-Level Repair.** If the maximal subcomponent repair fails (i.e., the scope has expanded to the full component without success), we escalate to repairing the FL component as a whole.

- **Statement-Level Repair.** As a final resort, if fine-grained repairs remain invalid, we fall back to repairing the entire FL statement.

To balance repair capabilities with computational efficiency, we limit each repair attempt at any granularity to a single inference pass. Furthermore, to ensure semantic correctness, we enforce the execution of statement-level repair as a mandatory semantic check, even after a successful local repair. The prompts for each level are detailed in Appendix D, and a concrete repair trajectory is visualized in Figure 4.

## 4. Experiments

### 4.1. The PRIME Benchmark

To evaluate the performance of DSR across a wide spectrum of mathematical complexity, we introduce the *Problem Repository of Instructional Mathematics for Evaluation (PRIME)*. Distinguishing it from prior benchmarks that focus on high school or undergraduate problems, PRIME comprises 156 theorem statements and proofs curated from both undergraduate and graduate-level textbooks, with the full list of sources detailed in Appendix C. As illustrated in Figure 5, the benchmark spans diverse domains including Algebra, Analysis, and Number Theory.

To ensure a high-fidelity ground truth, we adopted a meticulous construction process. For each entry, we extracted the NL statement and its informal proof from the source text. Subsequently, the NL statements were manually formalized by Lean experts into corresponding Lean 4 theorem statements. This expert-in-the-loop approach strictly enforces adherence to Lean's syntax and ensures semantic consistency between the informal and formal representations. Furthermore, the simultaneous extraction of natural language proofs extends PRIME's utility to the domain of Automated Theorem Proving (ATP).

### 4.2. Experiment Setting

**Benchmarks.** We employ a tiered suite of three benchmarks to assess model performance across varying levels of difficulty: (i) ProverBench (Ren et al., 2025), covering high school and introductory undergraduate problems; (ii) ProofNet (Azerbayev et al., 2023), the standard for undergraduate-level formalization; and (iii) PRIME (Section 4.1), extending the scope to advanced graduate-level theorems. Collectively, these datasets span diverse mathematical fields, ranging from Algebra and Number Theory to Real and Functional Analysis, ensuring a comprehen-

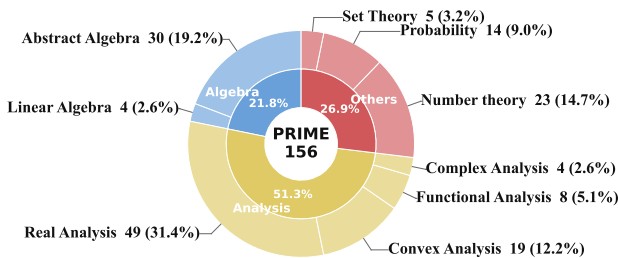

*Figure 5.* **Distribution of mathematical domains in PRIME**. The inner ring represents broad categories (Analysis, Algebra, Others), while the outer ring details specific sub-domains.

sive evaluation of general-purpose formalization capabilities rather than domain-specific proficiency.

**Baselines.** We benchmark DSR against a diverse set of state-of-the-art models, ranging from general-purpose reasoners like Qwen3-Max (Yang et al., 2025), which serves as the backbone for our repair module, to specialized formalizers including Kimina-Autoformalizer-7B (Wang et al., 2025a), StepFun-Formalizer-7B/32B (Wu et al., 2025), Goedel-V2-Formalizer-8B/32B (Lin et al., 2025), and ATF-8B/32B (Guo et al., 2025).

**Evaluations.** We employ a two-tiered protocol to assess formalization quality. First, we perform syntax checks using the Lean 4 compiler (toolchain `v4.15.0`) to verify compilability, reporting this metric as the **Syntax Check pass rate (SC)**. Second, we evaluate the semantic alignment between the NL statement and FL statement via LeanScorer (Yu et al., 2025a). Utilizing DeepSeek-V3.2 as the backbone with a recommended threshold of 0.6, we define this metric as the **Consistency Check pass rate (CC)**.

**Computational Budget.** To ensure an equitable comparison, we standardize the inference cost across all methods. Our analysis indicates that DSR requires an average of 2.9, 3.5, and 3.9 model calls on ProverBench, ProofNet, and PRIME, respectively. Based on this, we set a global budget of four inference calls per problem. This constraint is applied uniformly: as pass@4 for standard generation baselines and as a 4-turn iterative repair budget for repair-based baselines, including our ablation variants.

### 4.3. Experiment Results

The quantitative results across all three benchmarks are summarized in Table 2. Under the standardized computational budget of four inference calls (as defined in Section 4.2), DSR demonstrates consistently improvements over the baselines. Specifically, we derive the following key observations.

*Table 2.* **Overall results of DSR and competing baselines.** SC and CC denote Syntax Check and Consistency Check pass rates (%), respectively. All methods are restricted to a standardized computational budget of 4 inference calls. We evaluate baselines under two settings: standard sampling (*Pass@k, k = 4*) and a generic statement-level repair strategy (*Global Repair, N = 4*). *DSR-Global* denotes an ablation variant of our method utilizing only the generic statement-level repair instead of our proposed tree-guided strategy. The best results are presented in bold and the second highest with an underline.

| Model | ProverBench | | ProofNet | | PRIME | |
|---|---|---|---|---|---|---|
| | SC | CC | SC | CC | SC | CC |
| *Baselines (Pass@k, k = 4)* | | | | | | |
| Qwen3-Max | 82.15 | 68.62 | 74.12 | 62.80 | 63.46 | 56.41 |
| Kimina-Autoformalizer-7B | 93.23 | 65.54 | 83.02 | 56.87 | 75.00 | 48.08 |
| StepFun-Formalizer-7B | 76.92 | 57.54 | 54.18 | 43.40 | 47.44 | 42.31 |
| Goedel-V2-Formalizer-8B | 94.77 | 82.15 | 78.17 | 68.73 | 75.64 | 60.26 |
| ATF-8B-Distilled | 92.92 | 64.92 | 64.15 | 42.32 | 67.95 | 48.72 |
| StepFun-Formalizer-32B | 78.77 | 66.77 | 62.26 | 53.37 | 57.69 | 50.00 |
| Goedel-V2-Formalizer-32B | 95.38 | 83.38 | 77.63 | 70.89 | 77.56 | 66.67 |
| ATF-32B | 90.46 | 72.31 | 63.34 | 45.28 | 73.72 | 50.00 |
| *Baselines (Global Repair, N = 4)* | | | | | | |
| Qwen3-Max | 88.92 | 66.77 | 81.13 | 65.50 | 75.00 | 59.62 |
| Kimina-Autoformalizer-7B | 94.15 | 54.46 | 80.05 | 54.18 | 75.00 | 42.95 |
| StepFun-Formalizer-7B | 82.77 | 61.23 | 66.04 | 54.18 | 60.26 | 50.64 |
| Goedel-V2-Formalizer-8B | 95.69 | 72.92 | 73.58 | 60.92 | 73.72 | 54.49 |
| ATF-8B-Distilled | 88.92 | 42.46 | 66.04 | 38.54 | 67.95 | 35.26 |
| StepFun-Formalizer-32B | 81.85 | 64.92 | 70.08 | 56.60 | 67.95 | 54.49 |
| Goedel-V2-Formalizer-32B | **96.31** | 76.00 | 72.51 | 63.07 | 77.56 | 62.82 |
| ATF-32B | 90.77 | 50.46 | 67.92 | 39.35 | 74.36 | 37.18 |
| *Ours* | | | | | | |
| DSR-Global | 96.00 | 82.77 | **89.76** | 76.01 | **83.97** | 66.03 |
| **DSR** | 95.38 | **84.00** | 87.33 | **79.51** | 80.13 | **67.95** |

**Superiority in Formalization Quality.** DSR establishes a new state-of-the-art across all evaluated benchmarks, showcasing highly competitive performance. Despite the diversity of strong baselines, which range from specialized 7B models to the massive Qwen3-Max, DSR achieves the highest Consistency Check (CC) pass rates across the board. On ProverBench, ProofNet, and PRIME, it attains peak scores of 84.00%, 79.51%, and 67.95%, respectively. By doing so, DSR maintains a robust lead over the second-best performers, exemplified by a significant +8.62% margin over Goedel-V2-Formalizer-32B on ProofNet.

**Robustness on Complex Theorems.** Crucially, the performance advantage of DSR becomes increasingly pronounced as theorem complexity scales. While the performance gap on the entry-level ProverBench is relatively modest (0.62%), DSR maintains a steady lead on the graduate-level PRIME benchmark, outperforming the strongest baseline by 1.28%. This consistent trend suggests that the neuro-symbolic decomposition in DSR acts as a critical structural anchor. By explicitly mapping the hierarchical topology of theorems,

our framework robustly navigates intricate logical structures found in advanced mathematics.

**Fidelity in Semantic Alignment.** A critical weakness in some baselines is the discrepancy between syntactic validity and semantic correctness. For instance, on ProofNet, Kimina-Autoformalizer-7B achieves a high SC of 83.02% but drops significantly to 56.87% in CC. In contrast, models like Goedel-V2-Formalizer-32B and our DSR demonstrate exceptional semantic alignment with minimal SC-CC drops (e.g., 6.74% for Goedel and 7.82% for DSR). Notably, DSR distinguishes itself by achieving this high fidelity despite utilizing a significantly smaller underlying model (7B vs. 32B). This indicates that the OPT structure enforces strict semantic adherence to the natural language logic at advanced complexity levels.

### 4.4. Ablation Studies

In this section, we conduct comprehensive ablation studies to isolate the contributions of key components within DSR.

*Table 3.* **Ablation study of the DSR Formalizer training strategy.** We evaluate the incremental impact of incorporating operator trees and curriculum learning. The baseline denotes the standard training configuration that outputs only linear Lean code. Evaluations are performed across pass@k metrics where $k \in \{1, 4, 8\}$. The blue numbers indicate the absolute improvement over the baseline.

| Model | ProverBench | | ProofNet | | PRIME | |
|---|---|---|---|---|---|---|
| | SC | CC | SC | CC | SC | CC |
| *Pass@1* | | | | | | |
| Baseline | 29.54 | 21.54 | 15.09 | 12.13 | 22.44 | 18.59 |
| + Operator Tree | 31.69 (+2.15) | 24.92 (+3.38) | 15.63 (+0.54) | 11.86 | 19.87 | 16.03 |
| + Curriculum Learning | **32.62** (+3.08) | **25.85** (+4.31) | **18.87** (+3.78) | **16.44** (+4.31) | **23.08** (+0.64) | **20.51** (+1.92) |
| *Pass@4* | | | | | | |
| Baseline | 35.38 | 30.46 | 20.49 | 16.71 | 27.56 | 25.00 |
| + Operator Tree | 39.08 (+3.70) | 32.31 (+1.85) | **21.56** (+1.07) | 18.06 (+1.35) | 22.44 | 21.15 |
| + Curriculum Learning | **40.31** (+4.93) | **33.54** (+3.08) | 21.02 (+0.53) | **19.41** (+2.70) | **27.56** | **26.28** (+1.28) |
| *Pass@8* | | | | | | |
| Baseline | 37.54 | 32.92 | 21.83 | 18.60 | 28.85 | 26.28 |
| + Operator Tree | 40.31 (+2.77) | 35.38 (+2.46) | **23.45** (+1.62) | 19.41 (+0.81) | 23.08 | 23.08 |
| + Curriculum Learning | **42.15** (+4.61) | **36.92** (+4.00) | 23.18 (+1.35) | **21.02** (+2.42) | **29.49** (+0.64) | **27.56** (+1.28) |

Specifically, we analyze the impact of the proposed training strategies (Section 3.2) and validate the effectiveness of the tree-guided repair strategy (Section 3.3).

### 4.4.1. IMPACT OF THE TRAINING STRATEGY

To evaluate the efficacy of the proposed training strategy, we compare three incremental configurations: (1) **Baseline**: a standard sequence-to-sequence model mapping NL components directly to FL components; (2) **Baseline + Operator Tree**: the joint generation of FL components alongside FL OPTs; and (3) **Baseline + Curriculum Learning**: the complete framework incorporating the multi-stage training strategy detailed in Figure 3.

As summarized in Table 3, the introduction of operator tree supervision yields immediate gains on the ProverBench benchmark across all pass rates. However, on the more challenging PRIME benchmark, we observe an optimization barrier: merely adding the operator tree objective without a curriculum can lead to performance stagnation or even regression, where the Pass@1 SC drops from 22.44% to 19.87%. This suggests that simultaneously learning complex logic and hierarchical topology is non-trivial for the model.

Crucially, the integration of curriculum learning overcomes this barrier. By phasing the training difficulty, the model achieves significant improvements on PRIME, recovering to 23.08% Pass@1 SC, demonstrating that the phased strategy is essential for navigating high-complexity formalization.

Notably, this phenomenon is not unique to our component-based approach. As detailed in Appendix B.4, we observe identical trends in end-to-end models that map NL statements directly to FL statements, confirming that both incorporating operator tree supervision into training and employ-

ing curriculum learning are essential.

### 4.4.2. EFFECTIVENESS OF TREE-GUIDED REPAIR

We further investigate the effectiveness of the tree-guided repair strategy introduced in Section 3.3. To isolate the impact of our hierarchical approach, we compare DSR against external baselines and a DSR-Global ablation variant, where the latter utilizes the DSR Formalizer but relies solely on statement-level repair. Consistent with the equitable comparison framework established in Section 4.2, all methods are restricted to a standardized budget of $N = 4$ refinement attempts following the initial generation.

The results presented in the "Ours" section of Table 2 reveal a fundamental trade-off between syntactic validity and semantic fidelity. We observe that the DSR-Global variant occasionally achieves higher Syntax Check pass rates than the proposed DSR method. This phenomenon suggests that repairing the entire statement is an effective brute-force strategy for finding any compilable path. However, this syntactic success comes at the cost of semantic integrity, as evidenced by the consistently lower Consistency Check scores of the global variant. In contrast, our tree-guided approach performs surgical edits on localized errors to preserve the correct logic of the original generation.

Furthermore, DSR establishes a robust performance advantage over external baselines even when they are afforded the same computational budget of four refinement attempts. On the ProverBench dataset, our method remains competitive with Goedel-V2-Formalizer-32B. More importantly, on the complex PRIME benchmark, DSR outperforms the strongest baseline by a substantial margin in Consistency Check pass rates. These findings underscore that the efficacy of DSR stems not merely from iterative repairing but from

the structural precision of the operator tree. By reducing the repair scope to specific subcomponents, the framework minimizes the risk of introducing new errors while effectively correcting existing ones.

## 5. Conclusion

In this work, we present DSR, a novel neuro-symbolic framework that restructures autoformalization from a monolithic translation task into a modular pipeline of decomposition, structure, and repair. By explicitly modeling the hierarchical topology of decomposed components via operator trees, DSR effectively bridges the semantic gap between natural language and formal language. This structured representation not only significantly enhances the initial autoformalization capability but also provides a vital topological blueprint for precise, tree-guided repair, avoiding the semantic degradation inherent in global regeneration. Furthermore, we introduce the PRIME benchmark to rigorously evaluate formalization capabilities on advanced mathematics. Extensive experimental results demonstrate that DSR establishes a new state-of-the-art across multiple challenging datasets.

## Acknowledgements

This work is sponsored by the National Key R&D Program of China Grant No. 2022YFA1008200 (T. L.). We also thank Shanghai Institute for Mathematics and Interdisciplinary Sciences (SIMIS) for their financial support. This research was funded by SIMIS under grant number SIMIS-ID-2025-ST. The authors are grateful for the resources and facilities provided by SIMIS, which were essential for the completion of this work.

## Impact Statement

This work aims to advance autoformalization by decoupling the translation pipeline into distinct, structured stages, facilitating the formal verification of mathematics. By lowering the barrier to interactive theorem provers like Lean 4, frameworks like DSR help democratize formal verification for a broader community, enhancing the reliability of AI-assisted reasoning. However, deploying such automated systems introduces important considerations. A primary risk is automation bias, where users might overly trust a successfully compiled statement without verifying if the model preserved their original semantic intent. Furthermore, relying on LLMs for decomposition and repair raises concerns regarding computational overhead and equitable access.

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

# A. Limitations and Future Work

While the *Decompose, Structure, and Repair (DSR)* framework establishes a new state-of-the-art by modularizing the autoformalization process, we acknowledge that each stage introduces specific constraints that offer opportunities for future improvement:

- The success of DSR is heavily predicated on the initial semantic decomposition. While this approach effectively filters linguistic noise and explicates implicit context, it creates a potential bottleneck for error propagation: if the LLM fails to accurately partition the informal statement, the resulting operator tree will be inherently flawed from the outset.

- In our ablation studies on the PRIME benchmark, we observed that simply incorporating operator tree supervision alongside linear code generation occasionally led to a decrease in Consistency Check (CC) pass rates. This suggests a training bottleneck where the model struggles to simultaneously master complex mathematical logic and hierarchical topology. While our curriculum learning strategy mitigates this, future work should explore more effective ways to integrate OPTs during training.

- Through a preliminary analysis of the repair logs, we identified that a primary cause of formalization failure was the hallucination of non-existent identifiers or definitions within Mathlib. Consequently, future research should investigate more robust ways to integrate Retrieval-Augmented Generation (RAG) especially during the repair phase to effectively leverage domain knowledge without introducing additional semantic noise.

While this work currently prioritizes theorem statements, our future research will explore extending the DSR framework to the autoformalization of complete proofs. This trajectory ultimately leads toward the frontier of document-level formalization, a scale that introduces a foreseeable challenge: the necessity to handle mathematical concepts and definitions that transcend the current boundaries of Mathlib.

# B. Extended Experiment and Case Studies

### B.1. Analysis of Distributional Overlap in PRIME

To ensure the integrity of our evaluation and rule out any potential data contamination, we conducted a thorough investigation into the distributional overlap of the PRIME dataset. We clarify the following three aspects:

- **No DSR Training Overlap:** The training data for DSR spans from high school-level problems to synthetic undergraduate exercises. This ensures a strict domain separation from PRIME, which exclusively comprises advanced, graduate-level theorems.

- **No Prior Benchmark Overlap:** We cross-checked PRIME against widely used autoformalization benchmarks. Our analysis confirms that only 4 out of 156 problems share conceptual similarity with FATE-M, while the remaining 152 are entirely unique. Furthermore, even among these 4 conceptually similar problems, both the informal language (NL) and formal language (FL) representations differ significantly due to distinct formalization choices. For example:

---

**PRIME #133**
**NL:** If $(u, v) = 1$ and $uv = a^2$, show that both $u$ and $v$ are squares.
**FL:**
```
theorem prime_133 (u v a :  ℕ) (h1 :  Nat.Coprime u v)
   (h2 :  u * v = a^2) :  ∃ u1 v1 :  ℕ, u = u1^2 ∧ v = v1^2 := by sorry
```

**FATE-M #136**
**NL:** If $a$ and $b$ are positive integers with $(a, b) = 1$, and if $ab$ is a square, prove that both $a$ and $b$ are squares.
**FL:**
```
theorem fatem_136 {a b :  ℤ} (hab :  IsCoprime a b)
   (pos :  a > 0 ∧ b > 0) (hn :  IsSquare (a * b)) :
IsSquare a ∧ IsSquare b := by sorry
```

---

- **No Pretraining Leakage:** While the informal mathematical statements (textbook theorems) may inherently appear in the large-scale pretraining corpora of LLMs, their corresponding Lean code does not. The Lean code in PRIME were strictly manually formalized by our Lean experts and are entirely absent from any public repositories, thoroughly eliminating the risk of data memorization during base model pretraining.

## B.2. Empirical Validation of LeanScorer

To empirically validate the reliability of LeanScorer, we conduct a comprehensive manual evaluation across the entire PRIME dataset. Specifically, we humanly annotate all formalization outputs generated by both DSR and Goedel-V2-Formalizer-32B to establish a gold-standard human baseline. We then compare LeanScorer against a standard prompt-based LLM-as-a-Judge. Crucially, both the LLM-as-a-Judge and LeanScorer utilize DeepSeek-V3.2 as their underlying backbone. This controlled setup ensures that any performance discrepancy stems entirely from the architectural design of the evaluation metrics rather than variations in base model capacity.

**High Human Alignment.** As summarized in Table 4, LeanScorer significantly outperforms the standard LLM-as-a-Judge across precision, accuracy, and Cohen's Kappa. Notably, LeanScorer achieves a Kappa coefficient of 0.766, indicating substantial agreement with human experts, whereas the standard LLM judge only yields a moderate alignment of 0.603. This highlights the effectiveness of LeanScorer's specialized verification mechanism over naive prompt-based evaluation.

*Table 4.* Alignment comparison against human evaluation across the entire PRIME dataset.

| Metric | LLM-as-a-Judge (DeepSeek-V3.2) | LeanScorer (DeepSeek-V3.2) |
|---|---|---|
| Precision | 0.862 | **0.956** |
| Recall | **0.917** | 0.894 |
| Accuracy | 0.840 | **0.897** |
| Kappa | 0.603 | **0.766** |

**Cross-Model Stability.** To ensure the evaluation metric remains unbiased across different autoformalization models, we analyze its strictness across different model outputs. As shown in Table 5, LeanScorer maintains a consistent and proportional strictness gap relative to human judgments for both DSR and Goedel-V2-Formalizer-32B, confirming that it does not unfairly penalize any specific candidate model.

*Table 5.* Cross-model stability and strictness comparison on PRIME.

| Model | LLM-as-a-Judge | LeanScorer | Human |
|---|---|---|---|
| DSR | 76.92% | 67.95% | 71.15% |
| Goedel-V2-Formalizer-32B | 71.79% | 62.82% | 68.59% |

Furthermore, to understand why LeanScorer is overall slightly more rigorous than human experts (leading to a conservative pass rate), we conducted an in-depth error analysis on the discrepant cases. This investigation revealed two primary systematic causes:

1. **Isolated Condition Evaluation:** LeanScorer evaluates specific mathematical conditions in isolation. It occasionally misjudges complex scenarios where a single informal condition must be decoupled into multiple dependent Lean hypotheses, strictly flagging them as a "Major Inconsistency."

2. **Conservative Confidence Thresholding:** Valid formalizations that fall immediately below LeanScorer's strict, predefined confidence threshold are automatically rejected to heavily prioritize precision over recall.

## B.3. Performance on the FATE Series

To further evaluate the robustness of our proposed framework on formalizing complex mathematical problems with intricate cross-statement dependencies, we conduct supplementary experiments on the challenging FATE series (Jiang et al., 2026).

As shown in Table 6, DSR demonstrates highly competitive performance against strong baselines, including models with significantly larger parameter counts. We summarize the key observations as follows:

- **Superior Structural Accuracy:** DSR achieves the highest Syntax Check (SC) pass rates on both FATE-M (88.00%) and the highly complex FATE-X (31.00%). This underscores the effectiveness of our operator tree-guided strategy in ensuring the structural correctness of the generated formal specifications.

- **Highly Competitive Consistency:** For the Consistency Check (CC) metric, DSR consistently secures the second-highest performance across all three datasets (FATE-M, FATE-H, and FATE-X). It is only marginally outperformed by Goedel-V2-Formalizer-32B, a model with over four times the parameter count.

- **Robustness on Complex Contexts:** Notably, on FATE-X, which features the most challenging cross-line dependencies, DSR maintains its structural advantage and ties for the second-highest CC score. This highlights its superior capability to handle long-context mathematical reasoning compared to standard generation and global repair pipelines.

*Table 6.* **Evaluation results on the FATE series (FATE-M, FATE-H, FATE-X).** SC and CC denote Syntax Check and Consistency Check pass rates (%), respectively. All methods are restricted to a standardized computational budget of 4 inference calls. We evaluate baselines under two settings: standard sampling (*Pass@k*, $k = 4$) and a generic statement-level repair strategy (*Global Repair*, $N = 4$). The best results are presented in bold and the second highest with an underline.

| Model | FATE-M | | FATE-H | | FATE-X | |
|---|---|---|---|---|---|---|
| | SC | CC | SC | CC | SC | CC |
| *Baselines (Pass@k, $k = 4$)* | | | | | | |
| Qwen3-Max | 76.00 | 65.33 | 39.00 | 28.00 | 24.00 | 11.00 |
| Kimina-Autoformalizer-7B | 73.33 | 46.67 | 38.00 | 7.00 | 15.00 | 5.00 |
| StepFun-Formalizer-7B | 52.67 | 42.67 | 16.00 | 13.00 | 3.00 | 2.00 |
| Goedel-V2-Formalizer-8B | 86.67 | 70.00 | 58.00 | 37.00 | 26.00 | 10.00 |
| ATF-8B-Distilled | 58.00 | 44.67 | 24.00 | 12.00 | 6.00 | 2.00 |
| StepFun-Formalizer-32B | 58.67 | 48.00 | 18.00 | 16.00 | 8.00 | 6.00 |
| Goedel-V2-Formalizer-32B | 87.33 | **81.33** | **59.00** | **42.00** | 29.00 | **13.00** |
| ATF-32B | 63.33 | 51.33 | 26.00 | 16.00 | 8.00 | 3.00 |
| *Baselines (Global Repair, $N = 4$)* | | | | | | |
| Qwen3-Max | 83.33 | 58.00 | 48.00 | 27.00 | 30.00 | 10.00 |
| Kimina-Autoformalizer-7B | 62.67 | 32.00 | 32.00 | 8.00 | 19.00 | 6.00 |
| StepFun-Formalizer-7B | 54.67 | 36.00 | 20.00 | 8.00 | 10.00 | 6.00 |
| Goedel-V2-Formalizer-8B | 68.67 | 52.00 | 49.00 | 24.00 | 25.00 | 7.00 |
| ATF-8B-Distilled | 57.33 | 32.67 | 26.00 | 14.00 | 10.00 | 3.00 |
| StepFun-Formalizer-32B | 60.67 | 42.67 | 27.00 | 15.00 | 10.00 | 5.00 |
| Goedel-V2-Formalizer-32B | 81.33 | 66.67 | 51.00 | 34.00 | 26.00 | 10.00 |
| ATF-32B | 64.67 | 44.00 | 33.00 | 16.00 | 18.00 | 8.00 |
| *Ours* | | | | | | |
| **DSR** | **88.00** | 77.33 | 57.00 | 41.00 | **31.00** | 11.00 |

## B.4. Additional Ablation Studies

To further validate the efficacy of the operator tree and curriculum learning strategies introduced in Section 3.2, we conduct an additional ablation study using an end-to-end model.

**Experiment Setting.** Unlike the modular approach described in the main text, which operates on decomposed components, this model is trained to map complete NL statements directly to their corresponding FL statements.

- **Baseline**: A standard sequence-to-sequence training objective that maps NL statements directly to FL statements.

- **Baseline + Operator Tree**: The model is tasked with the joint generation of the FL statement and its associated FL OPT to encourage structural awareness during encoding.

- **Baseline + Curriculum Learning**: Building upon the second configuration, we incorporate a curriculum learning scheduler that transitions the model from simpler structural patterns to more complex formalizations.

**Results and Analysis.** Table 7 summarizes the performance across three benchmarks. On the ProverBench dataset, we observe that the integration of both operator tree and curriculum learning yields consistent improvements, demonstrating that structural supervision and staged training effectively guide the formalization process for simpler problems.

However, on more challenging benchmarks like ProofNet and PRIME, the efficacy of these strategies diminishes. While OPT supervision provides gains on ProofNet, yielding a 5.12% improvement in Pass@1 SC, it fails to improve, or even degrades, performance on the graduate-level PRIME benchmark. Furthermore, unlike in the component-based DSR, adding curriculum learning does not consistently resolve this optimization barrier in the end-to-end setting. We hypothesize that this plateau stems from the inherent complexity of OPTs derived from full mathematical statements. The excessive structural depth of a complete statement poses a significant optimization challenge, which dilutes the benefits of structural supervision.

These findings empirically justify our design choice in Section 3.2 to prioritize a component-level formalization strategy. By reducing the granularity of the task from full statements to localized components, we mitigate the learning difficulty associated with large-scale OPTs, thereby allowing the model to fully leverage the structural advantages.

*Table 7.* **Ablation study of the end-to-end model training strategy.** We evaluate the incremental impact of incorporating operator trees and curriculum learning into the end-to-end setting. The baseline denotes the standard training configuration that maps full NL statements directly to FL statements. Evaluations are performed across pass@$k$ metrics where $k \in \{1, 4, 8\}$. The blue numbers indicate the absolute improvement over the baseline.

| Model | ProverBench | | ProofNet | | PRIME | |
|---|---|---|---|---|---|---|
| | SC | CC | SC | CC | SC | CC |
| *Pass@1* | | | | | | |
| Baseline | 87.38 | 51.69 | 65.77 | 39.89 | 76.92 | 44.23 |
| + Operator Tree | 86.46 | 55.08 (+3.39) | **70.89** (+5.12) | **41.51** (+1.62) | 75.00 | 40.38 |
| + Curriculum Learning | **88.62** (+1.24) | **56.92** (+5.23) | 68.19 (+2.42) | 39.08 | 75.64 | **44.23** |
| *Pass@4* | | | | | | |
| Baseline | 92.92 | 72.00 | 77.63 | 53.64 | 84.62 | 53.21 |
| + Operator Tree | **93.85** (+0.93) | 72.92 (+0.92) | 82.48 (+4.85) | **56.60** (+2.96) | **85.26** (+0.64) | 53.21 |
| + Curriculum Learning | 92.92 | **74.77** (+2.77) | **83.02** (+5.39) | 53.64 | 83.97 | **54.49** (+1.28) |
| *Pass@8* | | | | | | |
| Baseline | 94.15 | 76.31 | 80.32 | 58.22 | 87.82 | 59.62 |
| + Operator Tree | **95.69** (+1.54) | 76.92 (+0.61) | 86.79 (+6.47) | **63.61** (+5.39) | 87.18 | 57.69 |
| + Curriculum Learning | 94.77 (+0.62) | **80.31** (+4.00) | **87.87** (+7.55) | 59.84 (+1.62) | **89.10** (+1.28) | 57.69 |

### B.5. Robustness and Error Mitigation in Decomposition

While the DSR pipeline inherently depends on the quality of the initial text decomposition (as discussed in Appendix A), we implement two key architectural measures to effectively mitigate the risk of sensitivity and error propagation:

- **High-Fidelity Initial Decomposition:** To minimize systemic errors at the source, we carefully selected the backbone model for this stage. Before finalizing the framework, we conducted a rigorous manual evaluation of 90 randomly sampled informal statements (30 each from ProverBench, ProofNet, and PRIME). Our analysis revealed that Gemini-3.0-Pro achieved a remarkable decomposition accuracy of 95.56%, significantly outperforming alternative models such as Qwen3-Max (87.78%). Deploying a highly capable model directly suppresses early-stage partitioning errors.

- **Global Semantic Verification via Repair:** Even if minor structural misalignments occur during the initial decomposition, they do not strictly dictate the final output. Our final statement-level repair stage explicitly revisits the original, complete informal statement as context. This design allows the system to identify and rectify any early partitioning biases during the feedback loop, guaranteeing that the final formalization maintains strict semantic fidelity to the original theorem.

### B.6. Inference Time Efficiency of DSR

To further clarify the computational cost of our framework, we recorded the precise inference times during our evaluation on the FATE series, as detailed in Table 8.

It is worth noting that the *decompose* and *structure* stages consume negligible time. The decomposition stage requires only a single API call for text parsing, while the structure stage (powered by our 7B model) generates an average of approximately 400 tokens (with a maximum limit of 2048 tokens). The primary time expenditure stems from interacting with the Lean compiler during the *repair* stage. This is because we have not yet applied engineering optimizations, and the repair phase currently operates sequentially in a single-item, single-threaded manner.

*Table 8.* Inference time statistics of DSR on the FATE series.

| Metric | FATE-M | FATE-H | FATE-X |
|---|---|---|---|
| Total Time (s) | 3329.413 | 3309.932 | 3870.774 |
| Average Time (s) | 22.196 | 33.010 | 38.708 |

## B.7. Case Study

To provide a granular understanding of the tree-guided repair strategy, this appendix further presents a series of repair cases.

---

**REPAIR EXAMPLE: ProofNet (`exercise_1_19c`)**

**NL Statement:** Prove that the power series $\sum_{n=1}^{\infty} \frac{z^n}{n}$ converges at every point on the unit circle except $z = 1$.

---

**Initial Generation State**

*NL Component:* $\forall z \in \mathbb{C}, \ (|z| = 1 \wedge z \neq 1) \implies \sum_{n=1}^{\infty} \frac{z^n}{n}$ converges.

*FL Component:* `∀ z : ℂ, (abs z = 1 ∧ z ≠ 1) → Summable fun n : ℕ => z^n / n`

**1. Subcomponent-Level Repair**

`(abs k = 1 ∧ z ≠ 1)`

✗ *Error: Function 'abs' not found in scope.*

↪ `(Complex.abs k = 1 ∧ z ≠ 1)`

✓ *Fix: Add namespace 'Complex'.*

**2. Statement-Level Repair**

`Summable fun n : ℕ => z^n / n`

✗ *Error: The series $\sum \frac{z^n}{n}$ is undefined at $n = 0$, causing a division by zero.*

↪ `Summable fun n : ℕ => z^(n + 1) / (n + 1)`

✓ *Fix: n + 1 can never be zero. Consistency check passed.*

---

**Final Verified Code:**

`theorem test: ∀ z : ℂ, (Complex.abs z = 1 ∧ z ≠ 1) → Summable fun n : ℕ => z^(n + 1)`
`/ (n + 1) := by sorry`

---

*Figure 6.* The model first resolves a namespace error by correcting `abs` to `Complex.abs` at the subcomponent level. It subsequently addresses a semantic inconsistency regarding a domain error by shifting the summation index to $n + 1$ at the statement level.

---

**REPAIR EXAMPLE: PRIME (`Exercise_1.3.3.1`)**

**NL Statement:** Let $g, h \in G$ be two commuting elements in a group $G$, with orders $o(g) = m$, $o(h) = n$, where $(m, n)$ denotes $\gcd(m, n)$ and $[m, n]$ denotes $\text{lcm}(m, n)$. Prove that

$$o(g^n h^m) = \frac{[m, n]}{(m, n)}.$$

---

**Initial Generation State**

| | | | |
|---|---|---|---|
| NL&FL Component 1: | $G$ is a group | $\longrightarrow$ | `[Group G]` |
| NL&FL Component 2: | $g, h \in G$ | $\longrightarrow$ | `(g h : G)` |
| NL&FL Component 3: | $m, n \in \mathbb{Z}^+$ | $\longrightarrow$ | `(m n : ℕ)` |
| NL&FL Component 4: | $gh = hg$ | $\longrightarrow$ | `g * h = h * g` |
| NL&FL Component 5: | $o(g) = m$ | $\longrightarrow$ | `o g = m` |
| NL&FL Component 6: | $o(h) = n$ | $\longrightarrow$ | `o h = n` |
| NL&FL Component 7: | $o(g^n h^m) = \frac{\text{lcm}(m,n)}{\gcd(m,n)}$ | $\longrightarrow$ | `o (g ^ n * h ^ m) = ...` |

**1. Component-Level Repair**

`(m n : ℤ⁺)`

× *Error: Nonnegative integers should be* $\mathbb{N}$*.*

↪ `(m n : ℕ)`

✓ *Fix: Replace* $\mathbb{Z}$*+ with* $\mathbb{N}$*.*

**2. Subcomponent-Level Repair**

`o g = m`

× *Error: 'Order' is not defined as* `o` *in Mathlib.*

↪ `o = m`

✓ *(Partial) Fix: remove application to satisfy parser/typechecker locally. (Semantically wrong; fixed globally later.)*

**3. Component-Level Repair**

`o h = n`

× *Error: 'Order' is not defined as* `o` *in Mathlib.*

↪ `orderOf h = n`

✓ *Fix: use* `orderOf` *for element order in Mathlib.*

**4. Component-Level Repair**

`o (g ^ n * h ^ m) = Nat.lcm m n / Nat.gcd m n`

× *Error: 'Order' is not defined as* `o` *in Mathlib.*

↪ `orderOf (g ^ n * h ^ m) = Nat.lcm m n / Nat.gcd m n`

✓ *Fix: replace the informal order notation* $o(\cdot)$ *by* `orderOf`*.*

**5. Statement-Level Repair**

`...(h2 : o = m) ...`

× *Issue: hypothesis became ill-formed semantically after local patch; it no longer states the order of g.*

↪ `(h2 : orderOf g = m)`

✓ *Fix: restore intended meaning* $o(g) = m$ *using* `orderOf g`*. Consistency check passed.*

---

**Final Verified Code:**

```
theorem test [Group G] (g h : G) (m n : ℕ) (h1 : g * h = h * g) (h2 : orderOf g = m)
(h3 : orderOf h = n) :  orderOf (g ^ n * h ^ m) = Nat.lcm m n / Nat.gcd m n := by
sorry
```

*Figure 7.* The repair starts by replacing the non-Lean type $\mathbb{Z}^+$ with $\mathbb{N}$. Then it repeatedly fixes the misuse of the informal order symbol `o` (treated as a numeral rather than a function) by switching to `orderOf`. A locally-compiling but semantically wrong patch `o = m` is later corrected at the global statement level into `orderOf g = m`. *Note: The declaration for type G is neglected during decomposition as this feature is normally absent in natural language, but Lean compiler automatically infers the type and inserts an implicit parameter to ensure syntactical correctness.*

**REPAIR EXAMPLE: ProofNet (`exercise_7_9`)**

**NL Statement:** Prove that a normal operator on a complex inner-product space is self-adjoint if and only if all its eigenvalues are real.

---

**Initial Generation State**
*NL Component:* $\forall V$ complex inner-product space,

$$\forall T \in \mathcal{L}(V), TT^* = T^*T \rightarrow (T = T^* \leftrightarrow \forall \lambda \in \mathbb{C}, (\exists v \in V, v \neq 0 \wedge Tv = \lambda v) \rightarrow \lambda \in \mathbb{R})$$

*FL Component:* `∀ (V : Type*) [InnerProductSpace ℂ V] (T : Module.End ℂ V),`
`T * T_star = T_star * T → (T = T_star ↔ ∀ (eigenvalue : ℂ),`
`(∃ (v : V), v ≠ 0 ∧ T v = eigenvalue · v) → eigenvalue ∈ Set.range (algebraMap ℂ ℝ))`

**1. Subcomponent-Level Repair**
`[InnerProductSpace ℂ V]`
× *Error: Failed to synthesize SeminormedAddCommGroup V.*
↪ `[SeminormedAddCommGroup V] [InnerProductSpace ℂ V]`
✓ *Fix: Add instance 'SeminormedAddCommGroup'.*

**2. Subcomponent-Level Repair**
`algebraMap ℂ ℝ`
× *Error: Failed to synthesize algebraMap ℂ ℝ.*
↪ `algebraMap ℝ ℂ`
✓ *Fix: ℂ is an algebra over ℝ, not the converse.*

**3. Subcomponent-Level Repair**
`T_star`
× *Error: T_star is undefined.*
↪ `T_star : Module.End ℂ V`
× *Failed: Error is caused by the definition of adjoint map, not the type of* `T_star`

**4. (Parent) Subcomponent-Level Repair**
`T = T_star ↔ ∀ (eigenvalue : ℂ), (∃ (v : V), v ≠ 0 ∧ T v = eigenvalue · v) →`
`eigenvalue ∈ Set.range (algebraMap ℝ ℂ)`
× Error: T_star is undefined.
↪ `T = star T ...`
× *Failed:* `star T` *is not the adjoint map in Mathlib.*

**5. (Parent) Component-Level Repair**
`∀ (V : Type*) [SeminormedAddCommGroup V] [InnerProductSpace ℂ V] (T : Module.End ℂ V),`
`T * T_star = T_star * T → (T = T_star ↔ ∀ (eigenvalue : ℂ), (∃ (v : V), v ≠ 0 ∧ T v`
`= eigenvalue · v) → eigenvalue ∈ Set.range (algebraMap ℝ ℂ))`
× *Error: T_star is undefined.*
↪ `... T = adjoint T ...`
× *Failed: The namespace* `LinearMap` *is omitted.*

**6. Statement-Level Repair**
`... T * T_star = T_star * T → (T = T_star ↔ ...)`
× *Error: T_star is undefined.*
↪ `...T * LinearMap.adjoint T = LinearMap.adjoint T * T → (T = LinearMap.adjoint T ↔ ...)`
✓ *Fix: Add the namespace and replace all* `T_star` *with* `LinearMap.adjoint T`*. Consistency check passed.*

---

**Final Verified Code:**
`theorem test : ∀ (V : Type*) [NormedAddCommGroup V] [InnerProductSpace ℂ V]`
`[FiniteDimensional ℂ V] (T : Module.End ℂ V), T * (LinearMap.adjoint T) =`
`(LinearMap.adjoint T) * T → (T = LinearMap.adjoint T ↔ ∀ (eigenvalue : ℂ), (∃ (v : V),`
`v ≠ 0 ∧ T v = eigenvalue · v) → eigenvalue ∈ Set.range (algebraMap ℝ ℂ)) := by sorry`

*Figure 8.* The model first corrects the instance issue and the incorrect definition at the subcomponent level, and then tries to resolve the undefined `T_star`. When this fails, it attempts to fix the issue in the parent scope, but still fails. Finally, it moves to the statement level and adds the missing namespace and the finite-dimensional instance.

# C. Benchmark Source

In particular, our benchmark PRIME is collected from the following mathematical textbooks:

- Rudin, W. *Principles of Mathematical Analysis* (3rd ed., 1976)

- Bauschke, H. H. & Combettes, P. L. *Convex Analysis and Monotone Operator Theory in Hilbert Spaces* (2nd ed., 2017)

- Ireland, K. & Rosen, M. *A Classical Introduction to Modern Number Theory* (2nd ed., 1990)

- Grimmett, G. & Stirzaker, D. *One Thousand Exercises in Probability* (3rd ed., 2020)

- Gelca, R. & Andreescu, T. *Putnam and Beyond* (2nd ed., 2017)

- De Souza, P. N. & Silva, J.-N. *Berkeley Problems in Mathematics* (3rd ed., 2004)

- Hungerford, T. W. *Algebra* (1974);

- Trench, W. F. *Introduction to Real Analysis* (2003)

- Chung, K. L. & AitSahlia, F. *Elementary Probability Theory With Stochastic Processes and an Introduction to Mathematical Finance* (4th ed., 2003)

- Alpay, D. *A Complex Analysis Problem Book* (2nd ed., 2016)

- Demidovich, B. P. *Problems in Mathematical Analysis* (1970)

- Feng, K. & Zhang, P. *300 Problems in Modern Algebra* (2009)

- Zhang, G. & Lin, Y. *Lecture Notes on Functional Analysis* (2nd ed., 2021)

- Xie, H. & Yun, Z. & Yi, F. & Qian, D. *Lecture Notes for Mathematical Analysis Problem Sessions* (2nd ed., 2018)

# D. Prompt Templates

---

**Prompt Template for Autoformalization (Qwen3-Max)**

You are an expert in mathematics and Lean 4.

Please autoformalize the following problem in Lean 4 with a header. Use the following theorem names:
my_favorite_theorem.
{informal_statement}

Your code should start with
```Lean4
import Mathlib
```

You should only output the theorem statement in Lean 4 format, ending with 'by sorry'. You should NOT output the proof.

---

**Prompt Template for Back-Translation**

# Role
You are a world−class expert in formal mathematics and the Lean 4 theorem prover.

# Input Data

---

∗∗Formal Conditions:∗∗ The (implicit/explicit) binders of the theorem in Lean 4.
∗∗Formal Conclusion:∗∗ The theorem statement in Lean 4.

# Task

Your goal is to translate the Formal Conditions one−by−one and Formal Conclusion back into natural language individually. Note that

1. Compared to written expressions, give priority to mathematical expressions (LaTeX format).
2. Translate each line as a whole, with each translation on a separate line ( −−> format). Do not add extra information or interpret beyond the given content.
3. Always translate the binders one−by−one, even if some of the binders can be merged in natural language. If the binder declares a variable (for example, 'a') to be a type, simply write 'Let a be a type'.
4. If the formal conclusion is a sequence of curried chain, you can start with 'If', and then stating the binders one−by−one in the curried chain, finally give the theorem statement.
5. If the input formal conditions are NULL, simply state the informal conditions as 'No conditions'.

# Output Format

Please present your response in the following structured format:
∗∗Informal Conditions:∗∗
∗∗Informal Conclusion:∗∗

# Example
# Input Data
∗∗Formal Conditions:∗∗
{a b c : $\mathbb{R}$}
(h : a ∗ b ∗ c = 1)
(haux : 1 + a + a ∗ b $\neq$ 0)

∗∗Formal Conclusion:∗∗
a / (a ∗ b + a + 1) + b / (b ∗ c + b + 1) + c / (c ∗ a + c + 1) = 1

# Expected Output
∗∗Informal Conditions:∗∗
{a b c : $\mathbb{R}$} −−> $a$, $b$, and $c$ are real numbers
(h : a ∗ b ∗ c = 1) −−> The product of $a$, $b$, and $c$ is equal to 1
(haux : 1 + a + a ∗ b $\neq$ 0) −−> The value $1 + a + ab$ is not equal to 0

∗∗Informal Conclusion:∗∗
a / (a ∗ b + a + 1) + b / (b ∗ c + b + 1) + c / (c ∗ a + c + 1) = 1 −−> The sum of the fractions $\frac{a}{ab + a + 1} + \frac{b}{bc + b + 1} + \frac{c}{ca + c + 1}$ is equal to 1

−−−

Now, perform the task for the following Input Data.

∗∗Formal Conditions:∗∗
{formal_conditions}

∗∗Formal Conclusion:∗∗
{formal_conclusion}

---

**Prompt Template for Decomposing Statements**

# Role
You are an expert mathematician and logic formalizer.

# Input Data
∗∗Problem Statement:∗∗ The problem statement in natural language.

# Task
Extract the Conditions (premises/givens) and Conclusions (goals/to−prove) from the mathematical problem statement.

IMPORTANT CONSTRAINTS:
1. Pure Formalization (No Solving): Express conditions and conclusions using concise LaTeX. Do not attempt to solve. Strip all redundant prose (e.g., "i.e.", "that is"). Prefer the shortest mathematically complete formulation.
2. Atomic Conditions: Each condition must contain exactly ONE atomic fact. Split compound statements into separate numbered lines for Lean compatibility.
3. Explicit Free Variables: Every free variable must be explicitly declared with its domain/type as a condition before it is used. If omitted in the text, infer the standard domain.
4. Implicit Structural Types: Expand structural relations into a type declaration plus a property condition. For example, "$A \subsetneq X$" MUST become two conditions: 1. "$A \subset X$" and 2. "$A \subsetneq X$".
5. Quantifier Strictness:
   − NEVER separate quantifiers from their predicates.
   − If the problem asks to "Find..." (existential) or asserts something for "any/every..." (universal), the entire statement MUST be a single quantified formula in the Conclusion.
   − DO NOT declare bound variables in the Conditions.
6. Empty Conditions: If the problem contains no independent premises, state: "No conditions."

# Output Format
∗∗Conditions:∗∗
1. ...
2. ...

∗∗Conclusion:∗∗
− ...

∗∗Important Note:∗∗ The ∗∗Conclusion∗∗ must be a ∗∗single line∗∗. Do NOT split the conclusion into multiple statements. All predicates and quantifiers must be combined into one formula.

−−−

Now, perform the task for the following Input Data.

∗∗Problem Statement:∗∗ {problem_statement}

---

**Prompt Template for Structuring Translation**

Please translate the natural language component into Lean4 code, and then parse it into a structured operator tree in JSON format. Use 'formal_content' for the operator logic (with '<SLOT>' as placeholders) and 'children' for the nested arguments.

Component: {text}
Tag: {tag}

**Prompt Template for Repairing Errors (Subcomponent Level)**

# Role
You are an expert in mathematics and Lean 4. You act as a "Micro−Surgeon" for Lean expressions, capable of fixing small fragments of code based purely on type constraints and compiler feedback.

# Input Data
**Broken Code:** A specific Lean 4 expression, term, or function call (a sub−segment of a line) containing errors.
**Error Message:** The raw error message (JSON−formatted) returned by the Lean 4 compiler.
**Previously Declared Variables:** A list of variables available in the local context (names and types).

# Note
Crucially, NO Informal Description is provided. You must infer the intended logic solely from the identifiers used in the 'Broken Code', the types of the available variables, and the specific error message.

# Task
Your goal is to fix the 'Broken Code' so that it passes type−checking when pasted back into its original position.
1. Scope Consistency (CRITICAL): The 'Broken Code' is a strict substring (an expression). Your output will be programmatically used to strictly replace it.
    − Output ONLY the expression. Do NOT add 'def', 'let', 'have', 'theorem', or assignment symbols (':=').
    − Do NOT output the surrounding code. If the input is 'MulAction.orbitRel G H', do not return '(h1 : Fintype ( MulAction.orbitRel G H))'.
2. Type−Driven Repair: Since there is no informal text, rely on Mathlib signatures and Type Theory:
    − Argument Order: Check if the function expects arguments in a different order.
    − Explicit/Implicit Arguments: Check if you need to make an argument explicit (using '@') or if you provided an explicit argument where an implicit one was expected.
    − Coercions: Check if a variable needs a conversion (e.g., 's' to 's.toFinset' or 'n' to '↑n').
    − Identifier Correction: If the error is "unknown identifier", find the correct existing Mathlib function name that closely matches the 'Broken Code'.
3. Analyze the Current Error: Examine the 'message' and 'position' to identify if the issue is a Type Mismatch, Unknown Identifier, or Synthesis Failure.
4. Check Context: Verify if the variables used are consistent with the 'Previously Declared Variables' (If any).
5. Apply Minimal Fixes: Correct the code only at the source of the error. Do not add any 'import' statements ( assume Mathlib is present).
6. Summarize: Write only a single sentence describing why the code failed (useful for classification).

# Output Format
Please present your response in the following structured format and do not include conversational filler.
**Error Reason:** <One−sentence summary, keep it as simple as possible>
**Corrected Code Snippet:** <The fixed expression ONLY.>

−−−

Now, perform the task for the following Input Data.

**Broken Code:** {broken_code}
**Error Message:** {error_message}
**Previously Declared Variables:** {previously_declared_variables}

**Prompt Template for Repairing Errors (Component Level)**

# Role
You are an expert in mathematics and Lean 4. You act as a "Code Surgeon" capable of fixing precise segments of code without disrupting the surrounding context.

# Input Data
**Informal Component:** The natural language or LaTeX description of the intended mathematics.
**Broken Code:** A snippet of Lean 4 code containing syntax or logical errors.
**Error Message:** The raw error message (JSON−formatted) returned by the Lean 4 compiler.
**Previously Declared Variables:** A list of variables available in the local context (If any).

# Task
Your goal is to fix the 'Broken Code' so that it compiles successfully when pasted back into the original context.
1. **Scope Consistency (CRITICAL):** The 'Broken Code' is a strictly defined substring of a larger file. Your output will be programmatically used to strictly replace the 'Broken Code' string.
    − Do NOT output the full theorem if the input was only a signature or a hypothesis.
    − Do NOT include surrounding keywords (like 'theorem', 'example', ':=', or 'by') unless they were strictly part of the 'Broken Code' string.
    − If you add context that wasn't in the input, the final concatenated code will fail (e.g., 'theorem theorem ...').
2. Semantic Alignment: Compare the 'Broken Code' against the 'Informal Component'. Ensure the fix preserves the intended logic for that specific context.
3. Analyze the Current Error: Examine the 'message' and 'position' in the Error Message to pinpoint the exact failure (e.g., incorrect syntax, type mismatch, unknown identifier).
    − If there is a type mismatch, check the 'Informal Component' to decide whether to cast/coerce variables or change the type definition.
4. Check Context: Verify if the variables used are consistent with the 'Previously Declared Variables' (If any).
5. Apply Minimal Fixes: Correct the code only at the source of the error. Do not add any 'import' statements ( assume Mathlib is present).
6. Summarize: Write only a single sentence describing why the code failed (useful for classification).

# Output Format
Please present your response in the following structured format and do not include conversational filler.
**Error Reason:** <One−sentence summary, keep it as simple as possible>
**Corrected Code Snippet:** <The fixed code snippet ONLY.>

−−−

Now, perform the task for the following Input Data.

**Informal Component:** {informal_component}
**Broken Code:** {broken_code}
**Error Message:** {error_message}
**Previously Declared Variables:** {previously_declared_variables}

**Prompt Template for Repairing Errors (Statement Level)**

# Role
You are an expert in mathematics and Lean 4. You act as both a SyntaxDebugger (fixing compilation errors) and a Semantic Auditor (ensuring faithfulness to the math).

# Input Data
**Informal Statement:** The natural language or LaTeX description of the mathematical proposition.
**Broken Statement:** The incorrect Lean 4 statement (theorem signature) containing syntax or logical errors.
**Error Message:** The raw error message (JSON−formatted) returned by the Lean 4 compiler.

# Task
Your goal is to ensure the 'Broken Statement' is both syntactically valid and semantically accurate.
1. **Analyze the Error Signal (CRITICAL BRANCHING):**
    **CASE A: 'Error Message' is PRESENT:**
        − Focus primarily on fixing the reported syntax or type error (e.g., "unknown identifier", "type mismatch").
        − Ensure the fix results in valid Lean 4 syntax.
    **CASE B: 'Error Message' is EMPTY/NULL:**
        − STOP DEBUGGING SYNTAX. The code already compiles.
        − Focus ONLY on Semantic Alignment. Compare the 'Broken Statement' strictly against the 'Informal Statement'.
        − Does it capture the correct mathematical meaning? Are there missing hypotheses? Is the formula correct?
        − If the statement is semantically correct, output it exactly as is.
        − Only modify the code if there is a clear logical deviation from the 'Informal Statement'.
2. Semantic Alignment: Compare the 'Broken Statement' against the 'Informal Statement'. Ensure the fixed code preserves the intended logic (quantifiers, implications, types) rather than just satisfying the compiler by changing the meaning.
3. Apply Minimal Fixes: Correct the code only at the source of the error. Do not add any 'import' statements ( assume Mathlib is present).
4. Summarize: Write only a single sentence describing why the code failed (useful for classification).

# Output Format
Please present your response in the following structured format and do not include conversational filler.
**Error Reason:** <One−sentence summary, keep it as simple as possible>
**Corrected Formal Statement:** <The fixed formal statement (theorem signature) only>

−−−

Now, perform the task for the following Input Data.

**Informal Statement:** {informal_statement}
**Broken Statement:** {broken_statement}
**Error Message:** {error_message}

