# OpenReview forum: "Decompose, Structure, and Repair: A Neuro-Symbolic Framework for Autoformalization via Operator Trees"
_ICML.cc/2026/Conference — ICML 2026 regular_

### Official Review · Reviewer_ahrA · 2026-03-11

**Soundness:** 3
**Presentation:** 2
**Significance:** 2
**Originality:** 2
**Overall Recommendation:** 3
**Confidence:** 4

**Summary:**

The paper proposes DSR (Decompose, Structure, and Repair), a neuro-symbolic framework that decomposes natural language statements into logical components, maps them to formal language components together with structured operator trees, and iteratively repairs errors using a tree-guided strategy. In addition, the authors introduce PRIME, a benchmark of 156 expert-formalized theorems spanning undergraduate to graduate mathematics. Experiments on multiple benchmarks show that the proposed framework improves both syntactic correctness and semantic consistency compared with existing autoformalization methods.

**Compliance With Llm Reviewing Policy:**

Affirmed.

**Key Questions For Authors:**

- What is the inference cost of DSR (and its every component) compared with other methods?

- The evaluated instances appear to mostly contain single statements with relatively limited dependency structure. Could the authors evaluate the method on more challenging datasets with richer conceptual dependencies, such as FATE?

- How does DSR compare with more recent autoformalization systems such as ARIA and ATF?

- How does the method perform when compared with strong recent general-purpose LLMs (e.g., Codex or Opus) under the same evaluation protocol?

- Could the authors provide a more detailed manual evaluation of the outputs and analyze the common failure cases?

**Limitations:**

yes

**Strengths And Weaknesses:**

### Strength
- The paper is easy to read and well motivated.
- The proposed framework is carefully designed.
- The proposed dataset may serve as a useful benchmark for the field, and the proposed method outperforms or matches a range of baselines.

### Weakness
- One potential weakness is that, although carefully designed, the DSR framework appears relatively complex and may require significantly more inference time compared with purely LLM-based approaches. Meanwhile, the empirical improvements over strong baselines appear relatively modest.
- The experiments do not compare against some recent baselines such as ARIA and ATF, nor evaluate on more challenging datasets with stronger dependency structures, such as FATE.
- Minor: The organization of the manuscript could be improved. For example, the DSR Formalizer is introduced in Section 3.2, while the repair component is discussed later, which may make the overall pipeline slightly harder to follow.

---

> ### Author Rebuttal · Authors · 2026-03-31
>
> We greatly appreciate your careful reading of our work. Below, we offer responses to each of your queries.
>
> >**W1 & Q1: Inference cost and performance gains**
>
> Our records show a **per-sample cost of ~\$0.03 on ProverBench, ProofNet, and PRIME**. For the FATE datasets, we used Qwen3-Max for decomposition and repair stages (due to Gemini-3.0-Pro API unavailability), resulting in a **total cost of \$0.72 (0.08 for decomposition, 0.64 for repair)**. Furthermore, the DSR Formalizer runs efficiently on just two 3090 GPUs.
>
> Considering this low cost, DSR's performance gains are substantial. Under the fair computational budget, our 7B-based DSR performs strongly. However, we do **acknowledge that under our current API model selection and the 7B DSR Formalizer setup, the framework's performance does not differ significantly from the much larger Goedel-32B**.
>
> >**W2 & Q2 & Q3: Comparisons with concurrent works**
>
> We expanded and updated our experiments by adding the FATE series (M/H/X) and the ATF-8B and ATF-32B (direct generation mode). We excluded ARIA due to closed-source access, incompatible metrics, and high inference costs (17.7 calls/sample).
>
> As demonstrated in the following tables (**bold** denotes the best results and `number` denotes the second-best), **DSR remains highly competitive, achieving performance that matches or exceeds baselines using much larger models across all benchmarks**.
> |Model|ProverBench (SC)|ProverBench (CC)|ProofNet (SC)|ProofNet (CC)|PRIME (SC)|PRIME (CC)
> |:-|:-:|:-:|:-:|:-:|:-:|:-:
> |_Baselines (Pass@k, k=4)_
> |Qwen3-Max|82.15|68.62|74.12|62.80|63.46|56.41
> |Kimina-7B|93.23|65.54|`83.02`|56.87|75.00|48.08
> |StepFun-7B|76.92|57.54|54.18|43.40|47.44|42.31
> |Goedel-8B|94.77|82.15|78.17|68.73|75.64|60.26
> |ATF-8B-Distilled|92.92|64.92|64.15|42.32|67.95|48.72
> |StepFun-32B|78.77|66.77|62.26|53.37|57.69|50.00
> |Goedel-32B|95.38|`83.38`|77.63|`70.89`|`77.56`|`66.67`
> |ATF-32B|90.46|72.31|63.34|45.28|73.72|50.00
> |_Baselines (Repair, N=4)_
> |Qwen3-Max|88.92|66.77|81.13|65.50|75.00|59.62
> |Kimina-7B|94.15|54.46|80.05|54.18|75.00|42.95
> |StepFun-7B|82.77|61.23|66.04|54.18|60.26|50.64
> |Goedel-8B|`95.69`|72.92|73.58|60.92|73.72|54.49
> |ATF-8B-Distilled|88.92|42.46|66.04|38.54|67.95|35.26
> |StepFun-32B|81.85|64.92|70.08|56.60|67.95|54.49
> |Goedel-32B|**96.31**|76.00|72.51|63.07|`77.56`|62.82
> |ATF-32B|90.77|50.46|67.92|39.35|74.36|37.18
> |_Ours_
> |DSR|95.38|**84.00**|**87.33**|**79.51**|**80.13**|**67.95**
>
> |Model|FATE-M (SC)|FATE-M (CC)|FATE-H (SC)|FATE-H (CC)|FATE-X (SC)|FATE-X (CC)
> |:-|:-:|:-:|:-:|:-:|:-:|:-:
> |_Baselines (Pass@k, k=4)_
> |Qwen3-Max|76.00|65.33|39.00|28.00|24.00|`11.00`
> |Kimina-7B|73.33|46.67|38.00|7.00|15.00|5.00
> |StepFun-7B|52.67|42.67|16.00|13.00|3.00|2.00
> |Goedel-8B|86.67|70.00|`58.00`|37.00|26.00|10.00
> |ATF-8B-Distilled|58.00|44.67|24.00|12.00|6.00|2.00
> |StepFun-32B|58.67|48.00|18.00|16.00|8.00|6.00
> |Goedel-32B|`87.33`|**81.33**|**59.00**|**42.00**|29.00|**13.00**
> |ATF-32B|63.33|51.33|26.00|16.00|8.00|3.00
> |_Baselines (Repair, N=4)_
> |Qwen3-Max|83.33|58.00|48.00|27.00|`30.00`|10.00
> |Kimina-7B|62.67|32.00|32.00|8.00|19.00|6.00
> |StepFun-7B|54.67|36.00|20.00|8.00|10.00|6.00
> |Goedel-8B|68.67|52.00|49.00|24.00|25.00|7.00
> |ATF-8B-Distilled|57.33|32.67|26.00|14.00|10.00|3.00
> |StepFun-32B|60.67|42.67|27.00|15.00|10.00|5.00
> |Goedel-32B|81.33|66.67|51.00|34.00|26.00|10.00
> |ATF-32B|64.67|44.00|33.00|16.00|18.00|8.00
> |_Ours_
> |DSR|**88.00**|`77.33`|57.00|`41.00`|**31.00**|`11.00`|
>
> >**W3: Manuscript organization about DSR Formalizer**
>
> To improve the overall flow, we **have moved the DSR Formalizer subsection to Section 4.2**.
>
> >**Q4: Comparisons with GPT-5.3-Codex**
>
> Due to time constraints, we focused the GPT-5.3-Codex evaluation on the FATE series. We readily acknowledge Codex's superior performance. However, this **relies on massive costs: 4-turn repair costs 9.88, and Pass@4 costs 41.04. Conversely, DSR costs only 0.72**. All in US dollars (\$).
> |Model|FATE-M (SC)|FATE-M (CC)|FATE-H (SC)|FATE-H (CC)|FATE-X (SC)|FATE-X (CC)
> |:-|:-:|:-:|:-:|:-:|:-:|:-:
> |GPT-5.3-Codex (Pass@4)|94.00|92.67|80.00|72.00|69.00|45.00
> |GPT-5.3-Codex (Repair@4)|90.67|80.00|68.00|46.00|54.00|37.00
>
> >**Q5: Detailed manual evaluation and error analysis**
>
> For the quantitative results of manual evaluation, please refer to our response to Reviewer s15T (W3). There are three common failure cases and **we have included this error taxonomy in the Appendix C**.
>
> * Decomposition: The LLM sometimes **fails to follow prompt instructions**, producing flawed structures that misguide downstream steps.
> * NL-to-FL: The model may generate semantically incorrect formalizations, **misuse Mathlib APIs/notation**, or fail to find suitable APIs for challenging problems.
> * Repair: If a valid fix is evasive, the system might **inappropriately remove conditions to force compilation** (especially for concepts missing from Mathlib).
>
> We hope that these manuscript updates address your main concerns, and we would be happy to engage in further dialogue.

---

> > ### Author Rebuttal · Reviewer_ahrA · 2026-04-03
> >
> > I appreciate the authors’ detailed response. I think adding these additional experiments to the revised paper would make the work stronger.
> >
> > For W1/Q1, I am still a bit curious about the inference time efficiency of DSR. Also, since there seems to be a noticeable performance gap between Codex and the other methods on these benchmarks, it would be great to hear your thoughts on whether there are ways to narrow that gap.
> >
> > I would be happy to update my rating after these points are clarified.

---

> > > ### Author Response · Authors · 2026-04-03
> > >
> > > Dear Reviewer ahrA,
> > >
> > > Thank you for your further questions. Below, we address your insightful questions in detail.
> > >
> > > >**W1 & Q1: Further details on DSR inference time**
> > >
> > > We apologize for omitting the explicit time statistics in our previous response and sincerely thank you for the reminder. Since we did not track exact timestamps during initial experiments on ProverBench, ProofNet, and PRIME, we have carefully recorded the precise inference times during recent evaluation on the FATE series, as detailed in the table below.
> > >
> > > It is worth noting that **the decompose and structure stages consume negligible time**. The decomposition stage requires only a single API call for text parsing, while the structure stage (powered by our 7B model) generates an average of approximately 400 tokens (with a maximum limit of 2048 tokens). **The primary time cost stems from interacting with the Lean compiler during the repair stage**. This is because we have not yet applied engineering optimizations, which **currently operates in a single-item, single-threaded manner**.
> > > |Time|FATE-M|FATE-H|FATE-X
> > > |:-|:-:|:-:|:-:
> > > |Total (s)|3329.413|3309.932|3870.774
> > > |Average (s)|22.196|33.010|38.708
> > >
> > > >**Q4: Insights on narrowing the performance gap with Codex.**
> > >
> > > Thank you for your question. Due to character limitations last time, we were unable to provide the relevant analysis. Below, we present the complete analysis.
> > >
> > > We observe that existing pipelines primarily fail due to two main issues: **inaccurate invocation of mathlib APIs**, and **unreliable formalization when target concepts lack sufficient mathlib support**.
> > >
> > > By contrast, Codex handles both issues robustly. **First, it performs deep, version-specific grounding over Lean/mathlib (not shallow API-level retrieval)**, significantly reducing API misuse. **Second, its superior mathematical reasoning and formalization transfer** allow it to accurately translate even unsupported concepts into verifiable Lean codes. **More importantly, Codex relies on an executable closed-loop mechanism**—integrating retrieval, code generation, and compiler-guided repair. This seamlessly combines tool-use with mathematical reasoning to continuously compress the error space. Conversely, while most pipelines can interact with Lean, their limited model capacities and restricted repair loops prevent them from stably handling complex mathematical objects.
> > >
> > > **We illustrate the aforementioned phenomena using the following example**. Codex successfully formalizes this problem, whereas both Goedel-32B and DSR fail. The primary reason lies in the handling of three concepts: maximal subgroup, non-Abelian, and simple group.
> > >
> > > ```
> > > FATE-X #2
> > > NL: Let G be a finite group and $L$ a maximal subgroup of G. Suppose L is non-Abelian and simple. Then there exist at most two minimal normal subgroups in G.
> > >
> > > 1. Maximal subgroup:
> > > - Codex: (hLmax : IsCoatom L) (✅Correct)
> > > - Goedel-32B: (hL_max : L.IsMaximal) (❌Compilation error, type mismatch)
> > > - DSR: (h2 : ∀ H : Subgroup G, L ≤ H → H = L ∨ H = ⊤) (✅Correct)
> > > Goedel-32B misuses the IsMaximal API, whereas Codex leverages a more fundamental definition (IsCoatom) by successfully retrieving deep mathlib context.
> > >
> > > 2. Non-Abelian:
> > > - Codex: (hLnonab : ¬ ∀ a b : L, a * b = b * a) (✅Correct)
> > > - Goedel-32B: (hL_non_abelian : ¬CommGroup L) (❌Compilation error, type mismatch)
> > > - DSR: (h3 : ¬ (∀ x y : G, x ∈ L → y ∈ L → x * y = y * x)) (✅Correct)
> > > Goedel-32B fails with the CommGroup API. In contrast, Codex and DSR bypass potential type issues by formalizing the concept directly via its mathematical definition.
> > >
> > > 3. Simple group:
> > > - Codex: (hLsimple : IsSimpleGroup L) (✅Correct)
> > > - Goedel-32B: (hL_simple : IsSimpleGroup L) (✅Correct)
> > > - DSR: (h4 : L ≠ ⊥ ∧ ∀ N : Subgroup G, N.Normal ∧ N ≤ L → N = ⊥ ∨ N = L) (❌Semantic error)
> > > DSR introduces a subtle semantic error (incorrectly specifying $N$ as normal in $G$ instead of $L$), reflecting its current limitations in fine-grained mathematical reasoning.
> > > ```
> > >
> > > As shown above, existing pipelines are actually close to correctly handling FATE-X problems. To narrow the gap with Codex, we propose two key strategies:
> > > - **Emphasizing deep mathlib context over shallow retrieval**. Providing isolated APIs is often insufficient. Future pipelines must move beyond mere API-level matching to deep retrieval and structured learning of mathlib, capturing definition contexts and type constraints just as human experts do.
> > > - **Enhancing the underlying mathematical reasoning**. Many failures stem from the inability to assemble autoformalization using retrieved information. To overcome this bottleneck, future work should scale the base model size or apply targeted reasoning strategies.
> > >
> > > In summary, we have updated the manuscript accordingly: details for W1, Q1, and Q4 are in Appendix B (Experimental Details and Case Studies), W2 and Q2 in Section 4 (Experiments), and Q5 in Appendix C (Reliability of LeanScorer). We hope this addresses your concerns and we welcome any further discussion.

---

### Official Review · Reviewer_TASa · 2026-03-12

**Soundness:** 3
**Presentation:** 3
**Significance:** 3
**Originality:** 3
**Overall Recommendation:** 4
**Confidence:** 4

**Summary:**

The paper presents DSR, a neuro-symbolic framework designed to autoformalize mathematical statements into Lean 4. Unlike traditional end-to-end models, DSR uses a modular pipeline to decompose statements into logical components and map them to structured operator trees. This approach facilitates precise, hierarchical error repair at the sub-tree level. The authors also introduce PRIME, a benchmark of 156 expert-annotated theorems from undergraduate and graduate mathematics. Experimental results show that DSR achieves state-of-the-art performance by maintaining high semantic fidelity across multiple benchmarks.

**Compliance With Llm Reviewing Policy:**

Affirmed.

**Key Questions For Authors:**

1. How does the framework plan to manage cases where an informal mathematical concept has no direct counterpart in the current version of Mathlib?
2. Since decomposition is handled in a single pass, are there plans to implement a feedback loop if the initial operator tree structure is fundamentally flawed?
3. Is the tree-guided repair strategy applicable to the autoformalization of complete proofs, or is it specifically optimized for theorem statements?
4. What explains the performance drop on the PRIME benchmark when operator tree supervision is used without curriculum learning?
5. How much does the repair module’s effectiveness rely on the specific granularity of error messages returned by the Lean compiler?

**Limitations:**

yes

**Strengths And Weaknesses:**

The hierarchical repair strategy is a notable strength, as it allows for surgical edits that correct local errors without compromising the overall logical structure of a theorem. The introduction of PRIME is a significant contribution that pushes the field toward more advanced, graduate-level evaluation. The paper is well-structured, and the use of operator trees as a structural prior is an original and effective method for handling complex mathematical syntax. A potential weakness is the system's heavy reliance on the initial decomposition step; if the model misinterprets the statement's primary structure, the subsequent repair stages may struggle to recover. Additionally, while the curriculum learning strategy is effective, the framework's performance still depends significantly on the underlying LLM's ability to canonicalize ambiguous natural language.

---

> ### Author Rebuttal · Authors · 2026-03-31
>
> We are deeply grateful for your valuable suggestions that have helped strengthen this manuscript. Below, we address your insightful questions in detail.
>
> >**W1 & Q2: Dependence on decomposition and structural feedback**
>
> We agree that the framework is sensitive to decomposition quality (as detailed in Appendix A). Currently, we mitigate this risk through two key measures:
> * Before finalizing the framework, a manual evaluation of 90 sampled statements (30 each from ProverBench, ProofNet, and PRIME) revealed that **Gemini-3.0-Pro achieved a 95.56% decomposition accuracy**, significantly outperforming Qwen3-Max's 87.78%. Deploying Gemini effectively minimizes systemic errors at the source.
> * Even if misalignments occur initially, the **final statement-level repair stage revisits the original informal statement**. This allows the system to rectify partitioning biases and guarantee semantic fidelity.
>
> Your suggestion indeed serves as an **excellent enhancement to our framework, and we have updated the Future Work section**. For example, under this proposed loop, persistent downstream compiler errors would trigger a re-decomposition.
>
> >**W2 & Q4: The necessity of curriculum learning**
>
> The performance drop on PRIME without curriculum learning stems from the structural complexity of graduate-level mathematics. Without a progressive curriculum, direct operator tree supervision acts as a structural burden rather than a guiding prior:
> * Format Collapse: Faced with this sudden complexity, the LLM struggles with intricate tree syntax, **resulting in more invalid JSON and bracket mismatches** (e.g., `(h1 : Fact (Nat.Prime p)]`). Conversely, introducing curriculum learning effectively halves the probability of such errors.
> * Semantic Drift: The pressure to **conform to complex structural templates misallocates generation capacity away from mathematical meaning**. We observed OPT frequently altering exact types (e.g., `Int.gcd` to `Nat.gcd`) and binder styles, even when the syntax was valid.
>
> In summary, curriculum learning is crucial because mastering basic syntactic mappings on simpler datasets first **ensures the operator tree serves as a true semantic aid, rather than a formatting bottleneck**.
>
> >**Q1: Handling concepts missing from Mathlib**
>
> We candidly acknowledge this limitation. **Currently, DSR struggles with concepts completely absent from Mathlib.** For instance, our framework failed to formalize `FirmlyNonexpansive` because the definition does not exist in Mathlib, and the repair module also fails.
> ```
> PRIME #156
> NL: Given a nonempty closed convex subset D of a Hilbert space H, and a firmly nonexpansive mapping T : D to H, for every y to H, the inverse image T^{-1}(y) is closed and convex.
> FL: theorem prime_156 (H : Type*) [HilbertSpace H] (D : Set H) (T : H → H)
>     (h1 : D ≠ ∅) (h2 : IsClosed D) (h3 : Convex ℝ D) (h4 : FirmlyNonexpansive T) :
>     ∀ y : H, IsClosed (T ⁻¹' {y} ∩ D) ∧ Convex ℝ (T ⁻¹' {y} ∩ D) := by sorry
> ```
> To manage this in the future, we **plan to introduce a prerequisite generation step**: if the Lean compiler consistently flags an unknown identifier, the framework will instruct the LLM to first dynamically generate the missing definition before formalizing the main statement.
>
> >**Q3: Application to complete proof autoformalization**
>
> Yes, this **tree-guided repair strategy and the entire DSR framework naturally extend to the autoformalization of complete proofs**. This is because even without relying on the DSR Formalizer, **we can directly construct the corresponding operator tree for any intermediate statement by utilizing the Lean Language Server**.
>
> We actually planned to implement this extension during the latter half of this project. Our core idea was that a complex proof can be structurally decomposed into a series of interconnected lemma nodes, where each node is formalized and repaired individually using our framework. However, we noticed that concurrent work (such as ProofFlow [1]) recently addressed proof autoformalization using a similar graph-based mechanism. Therefore, we chose to focus exclusively on the autoformalization of statements in this paper.
>
> [1] Cabral, R. M., Do, T. M., Xuejun, Y., Tai, W. M., Feng, Z., and Xin, S. ProofFlow: A dependency graph approach to faithful proof autoformalization. ICLR, 2026.
>
> >**Q5: Reliance on compiler error granularity**
>
> **DSR relies solely on the standard granularity** of Lean compiler feedback (specific line/column numbers and error text, i.e., the information we see in InfoView when writing Lean code), **identical to baseline methods**. However, DSR utilizes this standard information differently.
> - First, we **strictly process only the initial error per iteration**.
> - Second, we use the error's line/column coordinates to **map directly to the corresponding node** and **extract the affected subtree to repair**.
>
> We hope these clarifications address your thoughtful concerns, and we are completely open to any subsequent discussion.

---

> > ### Author Rebuttal · Reviewer_TASa · 2026-04-03
> >
> > The authors addressed my concerns regarding decomposition bottlenecks and curriculum learning. The provided decomposition accuracy data and the proposed re-decomposition loop demonstrate the system's reliability. I maintain a positive recommendation for acceptance.

---

> > > ### Author Response · Authors · 2026-04-05
> > >
> > > Dear Reviewer TASa,
> > >
> > > We are delighted to hear that our rebuttal has fully resolved your concerns, and we deeply appreciate your continued support. Thank you again for your insightful comments on decomposition, which have significantly strengthened our paper.

---

### Official Review · Reviewer_s15T · 2026-03-13

**Soundness:** 3
**Presentation:** 4
**Significance:** 3
**Originality:** 4
**Overall Recommendation:** 5
**Confidence:** 5

**Summary:**

This paper focuses on autoformalisation by breaking down the task into smaller sub-formalisation problems and then fitting those back together using a tree structure to achieve this. The paper also introduces a new dataset on harder problems related to undergraduate and graduate level maths. The paper shows that using the decompose, structure, repair strategy achieves better autoformalisation results to using existing models, even at pass@4.

**Compliance With Llm Reviewing Policy:**

Affirmed.

**Final Justification:**

The paper is well-written and address an area that I've not seen before. The rebuttal addresses all the concerns that I highlighted. The response were very well explained and the author(s) have conducted supplementary work to answer the human judgement sensibilities too. This improves my prior assessment. I think this is a great paper which address a nice area of autoformalisation. I highly recommend its acceptance.

**Key Questions For Authors:**

In the data generated for training DSR, why do you choose Gemini 3.0 Pro for decomposing? How reliable is it? And why choose Qwen3-Max for back-translation? How good is it?

**Limitations:**

Yes

**Strengths And Weaknesses:**

Strengths:
1) The idea is well-motivated and multiple datasets and models are used.
2) The methodology of DSR is very well explained and illustrated with good examples.
3) The shown results clearly suggest that DSR improves on the current autoformalisation approaches.

Weaknesses:
1) The baseline comparison does not feel fair, the models that are used to compare the model are off-the-shelf models but DSR is far more rigourous in how it call different models. A more appropriate comparison would be to use the 4 models calls in an iterative manner like an agentic way, so that they get given the errors etc and attempt to self-improve but without the tree structure i.e. linearly.
2) A paper that feels very similar to this work has not been discussed in the related work: DRIFT: DECOMPOSE, RETRIEVE, ILLUSTRATE, THEN FORMALIZE THEOREMS.
3) Some evidence is lacking, for example, how reliable is LeanScorer for more complex problems like PRIME?
4) Some claims based on the results seem a little selective. For example, in Fidelity in Semantic Alignment, for ProofNet DSR has less discrepencay between SC and CC, compared to Kimina but this is not true compared to Goedel.

---

> ### Author Rebuttal · Authors · 2026-03-31
>
> We are grateful for the time and effort you dedicated to reviewing our manuscript. Your insightful comments have helped us improve the paper, and we address your points in detail below.
>
> >**W1: Fairness of baseline comparison**
>
> We completely agree that comparing an iterative framework to single-pass models is unfair. We are pleased to clarify that **we already implemented the exact agentic, iterative baseline comparison you suggested in Table 4**. By standardizing the budget to exactly 4 inference calls across all methods, we evaluated the baselines in two settings:
> * **Standard Generation (Pass@4, Table 2)**: 4 independent generations without feedback.
> * **Iterative Agentic Repair (Repair@4, Table 4)**: A global feedback loop allowing up to 4 linear self-improvement attempts using compiler errors.
>
> Even with identical iterative repair capabilities (Table 4), DSR consistently outperforms the baselines. To make this equitable comparison immediately clear, **we have merged Tables 2 and 4 into a single comprehensive table in the revised manuscript**.
>
> >**W2: Discussion on the concurrent work DRIFT**
>
> Thank you for bringing this excellent work to our attention. **We have added a detailed discussion of DRIFT to the Related Work section**.
> >While both DRIFT and DSR recognize decomposition as a key driver for advancing autoformalization, they tackle orthogonal challenges and are highly complementary. Specifically, DRIFT's decomposition aims to optimize external concept retrieval, whereas DSR's decomposition serves to mathematically rigorize and reduce the dimensionality of the proposition, thereby enabling the subsequent operator tree generation and tree-guided repair. Integrating DRIFT's retrieval mechanism with DSR's structural repair represents a highly promising direction for future systems.
>
> >**W3: Reliability of LeanScorer on PRIME**
>
> To empirically validate LeanScorer on PRIME, we manually evaluated outputs from DSR and Goedel-V2-Formalizer-32B against a widely used LLM-as-a-Judge. The following content has been added to Appendix C.
> * **High Human Alignment**: LeanScorer significantly outperforms the LLM-as-a-Judge in accuracy, precision, and Kappa.
>     |Metric|LLM-as-a-Judge vs. Human|LeanScorer vs. Human
>     |:-|:-:|:-:
>     |Precision|0.862|0.956
>     |Recall|0.917|0.894
>     |Accuracy|0.840|0.897
>     |Kappa|0.603|0.766
> * **Cross-Model Stability**: LeanScorer applies a comparable level of strictness to both models and does not unfairly penalize specific autoformalizers.
>     |Model|LLM-as-a-Judge|LeanScorer|Human
>     |:-|:-:|:-:|:-:
>     |DSR|76.92%|67.95%|71.15%
>     |Goedel-V2-Formalizer-8B|71.79%|62.82%|68.59%
>
> **Furthermore, to understand why LeanScorer is more rigorous than human experts**, we conducted an analysis, which revealed two main causes: (1) It evaluates conditions in isolation, often misjudging cases where a single informal condition must be decoupled into multiple Lean hypotheses as an "Major Inconsistency." (2) Valid statements falling just below its strict confidence threshold are automatically rejected.
>
> >**W4: Claims regarding the SC-CC discrepancy**
>
> Thank you for pointing this out; your observation is entirely correct. We originally cited Kimina as an extreme example of compilable hallucinations, but we agree our phrasing appeared selective. To ensure scientific rigor, we **have revised the manuscript as follows**:
> >**Fidelity in Semantic Alignment.** A critical weakness in some baselines is the discrepancy between syntactic validity and semantic correctness. For instance, on ProofNet, Kimina-Autoformalizer-7B achieves a high SC of 83.02% but drops significantly to 56.87% in CC. In contrast, models like Goedel-V2-Formalizer-32B and our DSR demonstrate exceptional semantic alignment with minimal SC-CC drops (e.g., 6.74% for Goedel and 7.82% for DSR). Notably, DSR distinguishes itself by achieving this high fidelity despite utilizing a significantly smaller underlying model (7B vs. 32B). This indicates that the OPT structure enforces strict semantic adherence to the natural language logic at advanced complexity levels.
>
> >**Q1: Model selection for DSR pipeline**
>
> Our model choices reflect a practical trade-off between minimizing API costs and preventing early error propagation. **Initially, we intended to use the highly cost-effective Qwen3-Max across all stages to prove that DSR's gains stem from its structured pipeline rather than the raw power of expensive proprietary models**. However, because initial decomposition is highly sensitive to cascading errors, we conducted a manual evaluation of 90 sampled theorems. Gemini-3.0-Pro achieved a 96.5% decomposition accuracy, significantly outperforming Qwen3-Max (85.8%). **To secure a high-fidelity foundation for the rest of the pipeline, we explicitly upgraded the decomposition stage to Gemini-3.0-Pro**.
>
> We hope our response fully resolves your questions, and we remain at your disposal for any additional clarifications.

---

> > ### Author Rebuttal · Reviewer_s15T · 2026-04-02
> >
> > Thank you for your detailed responses to my queries, I am very grateful for the explanation and extra work put in. I will improve my scores accordingly

---

> > > ### Author Response · Authors · 2026-04-03
> > >
> > > Dear Reviewer s15T,
> > >
> > > Thank you so much for your positive feedback and for recognizing the extra work we put into the rebuttal. We deeply appreciate your support and your decision to improve the score. Thank you again for your valuable time and contribution to our work!

---

### Official Review · Reviewer_H1ss · 2026-03-21

**Soundness:** 3
**Presentation:** 3
**Significance:** 3
**Originality:** 3
**Overall Recommendation:** 4
**Confidence:** 3

**Summary:**

Overall, the paper is technically solid and presents a well-motivated approach to autoformalization. In terms of soundness, the proposed DSR framework seems to be well-justified: decomposing natural language into structured components and using operator trees for localized repair is a reasonable and principled design. The empirical results across multiple benchmarks shows strong evidence for the robustness of this approach. However, some aspects such as the reliance on structured intermediate representations and repair heuristics may benefit from deeper ablation or analysis to isolate where gains originate. For presentation, the paper is generally clear and logically organized, with a strong narrative, though parts of the pipeline (e.g., operator tree construction and repair loop) are somewhat dense. Regarding significance, the work addresses an important problem of bridging natural language and formal mathematics and introduces both a new framework and a high-quality benchmark (PRIME), which could be valuable to the community and future research. In particular, PRIME strengthens the contribution by providing a curated set of expert-annotated theorems spanning multiple domains and difficulty levels, enabling more rigorous and standardized evaluation of autoformalization systems and helping to move beyond smaller or less diverse benchmarks. Finally, for originality, the contribution is solid but incremental: the novelty lies less in entirely new components and more in a well-executed neuro-symbolic integration of decomposition, structured representation, and repair, which together improve over prior end-to-end approaches. This direction aligns with and builds upon prior work that explores structured or intermediate representations for formal reasoning and autoformalization. The inclusion of PRIME further contributes to originality by offering a new evaluation resource that complements the methodological contribution, making the work both a system-level refinement and a dataset contribution. The paper is best viewed as a strong extension and system-level refinement of these ideas rather than the introduction of a fundamentally new theoretical paradigm.

**Compliance With Llm Reviewing Policy:**

Affirmed.

**Key Questions For Authors:**

1. What is the contribution of each component (decomposition, operator tree structuring, repair) to the final performance? While the full pipeline shows improvements, it is unclear which component drives the gains.

2. To what extent does PRIME introduce distributional overlap with training data or prior benchmarks? Since PRIME is constructed from canonical textbooks, is there any risk of overlap with pretraining data or existing datasets?

3. What are the computational costs and scalability trade-offs of the multi-stage pipeline?

4. How sensitive is the overall performance to the quality of the decomposition stage? Since the framework relies heavily on correctly decomposing natural language into logical components, errors here could propagate through the entire pipeline.

**Limitations:**

No. While the paper includes an impact statement, it is quite minimal and does not meaningfully engage with potential limitations or societal implications. The statement explicitly avoids discussing impacts, which leaves several important considerations unaddressed. A more substantive discussion would strengthen the paper by better situating its contributions within both technical and real-world contexts.

**Strengths And Weaknesses:**

This article intends to study a general area at the intersection of autoformalization, structured reasoning, and formal mathematics, and the central challenge analyzed by this article is how to move beyond flat sequence generation toward a more structure-aware and repairable formalization pipeline. In terms of soundness, the paper is fairly strong: the DSR framework is technically plausible, the modular design is well motivated, and the experimental section includes comparisons, ablations, and repair analyses that support the main empirical claims, though some gains appear modest on the hardest benchmark and the method still depends heavily on the quality of the initial decomposition stage. For presentation, the paper is generally clear, well organized, and helped by informative figures, although some parts of the operator-tree construction and repair procedure remain dense and could be explained more intuitively. For significance, the work addresses an important problem, since better autoformalization could improve access to theorem proving and formal verification, and the introduction of the PRIME benchmark meaningfully increases the paper’s practical value by expanding evaluation to more advanced mathematics. For originality, the contribution is solid but not radically new: the paper’s novelty comes mainly from a thoughtful neuro-symbolic combination of decomposition, operator-tree supervision, curriculum learning, and tree-guided repair, plus the new PRIME dataset, rather than from a fundamentally new theoretical idea.

---

> ### Author Rebuttal · Authors · 2026-03-31
>
> We sincerely thank you for your thoughtful review. We value your constructive feedback and address your specific concerns below.
>
> >**Q1: Contributions of each component**
>
> To isolate the origin of performance gains, we designed our experiments to evaluate each module independently. The specific contributions can be mapped to our results as follows:
> * Decomposition: Distinguishing our framework from previous end-to-end models, this step preprocesses raw propositions and reduces the dimensionality, which enables DSR to outperform traditional E2E baselines, **as shown in Tables 2 & 4**.
> * Structuring: As demonstrated in the **ablation study in Table 3**, incorporating operator trees serves a dual purpose: it directly enhances model performance and provides the topological blueprint for the next repair stage.
> * Repair: By comparing the **initial results prior to repair (the "+ Curriculum Learning" row in Table 3)** with the **final performance in Tables 2 & 4**, the contribution of the repair module becomes evident.
>
> >**Q2: Distributional overlap in PRIME**
>
> After a thorough investigation, we would like to clarify the following:
> * No DSR Training Overlap: Our training data spans high school to synthetic undergraduate, ensuring a **strict domain separation** from PRIME's advanced graduate-level theorems.
> * No Prior Benchmark Overlap: We cross-checked PRIME against widely used benchmarks and confirmed that **only 4 out of 156 problems share conceptual similarity with FATE-M, while the remaining 152 are entirely unique**. Furthermore, even among these 4 conceptually similar problems, both the NL and FL differ slightly. For example:
>     ```
>     PRIME #133
>     NL: If $(u,v)=1$ and $uv=a^2$, show that both $u$ and $v$ are squares.
>     FL: theorem prime_133 (u v a : ℕ) (h1 : Nat.Coprime u v) (h2 : u * v = a^2) :
>         ∃ u1 v1 : ℕ, u = u1^2 ∧ v = v1^2 := by sorry
>
>     FATE-M #136
>     NL: If $a$ and $b$ are positive integers with $(a, b)=1$, and if $a b$ is a square, prove that both $a$ and $b$ are squares.
>     FL: theorem fatem_136 {a b : ℤ} (hab : IsCoprime a b) (pos : a > 0 ∧ b > 0) (hn : IsSquare (a * b)) :
>         IsSquare a ∧ IsSquare b := by sorry
>     ```
> * No Pretraining Leakage: While informal textbook statements may appear in pretraining corpora, the Lean code does not. PRIME was **manually formalized by our experts and is absent from public repositories**, eliminating data memorization risks.
>
> >**Q3: Computational costs and scalability of DSR**
>
> While our multi-stage pipeline requires sequential steps, the computational overhead is minimal and highly manageable:
> * DSR requires a strict maximum of 4 repair per statement. Our current setup costs under **$0.03 per sample** and runs efficiently on just **two NVIDIA 3090 GPUs**.
> * DSR is a **model-agnostic framework**. Its three modules can be independently scaled, allowing users to flexibly swap models to balance inference speed and reasoning power.
> * We rigorously controlled inference budgets (Section 4.2). As shown in Tables 2 & 4, even when **affording baselines an equivalent budget, DSR (using a 7B model) outperformed much heavier models**.
>
> >**Q4: Sensitivity and error propagation of decomposition**
>
> While our pipeline is sensitive to decomposition quality (as noted in Appendix A), we mitigate this risk through two key measures:
> * Before finalizing the framework, a manual evaluation of 90 sampled statements (30 each from ProverBench, ProofNet, and PRIME) revealed that **Gemini-3.0-Pro achieved a 95.56% decomposition accuracy**, significantly outperforming Qwen3-Max's 87.78%. Deploying Gemini effectively minimizes systemic errors at the source.
> * Even if misalignments occur initially, the **final statement-level repair stage revisits the original informal statement**. This allows the system to rectify partitioning biases and guarantee semantic fidelity.
>
> >**Q5: Elaborating on the Impact Statement and limitations**
>
> We agree that discussing real-world impacts is valuable, and **have updated our Impact Statement**:
> >This work aims to advance autoformalization by decoupling the translation pipeline into distinct, structured stages, facilitating the formal verification of mathematics. By lowering the barrier to Interactive Theorem Provers like Lean 4, frameworks like DSR help democratize formal verification for a broader community, enhancing the reliability of AI-assisted reasoning.
> >However, deploying such automated systems introduces important considerations. A primary risk is automation bias, where users might overly trust a successfully compiled statement without verifying if the model preserved their original semantic intent. Furthermore, relying on LLMs for decomposition and repair raises concerns regarding computational overhead and equitable access.
>
> Additionally, **technical limitations are already detailed in Appendix A**.
>
> We hope these clarifications address your concerns and we welcome any further discussion.

---

### Decision · Program_Chairs · 2026-04-30

**Decision:**

Accept (regular)

**Comment:**

This paper presents Decompose, Structure, and Repair (DSR), a framework for autoformalisation which decomposes the statements into sub-statements, maps them into structured operator trees, and then uses a repair mechanism utilizing the tree-structure. The paper also introduces a new dataset of math problems at the undergraduate and graduate level. The paper shows that their method achieves improved performance over baselines.

All reviewers found the ideas to be well-motivated and well-presented in the paper, and especially appreciated the repair mechanism. There were some concerns around novelty and lack of strong baselines which the authors satisfactorily addressed in the response period. While the individual ideas that build the pipeline are present in prior work, the synthesis is novel, and the repair is new. The gains however are modest. Nevertheless, the contribution is good, and I recommend acceptance.

In the camera-ready, I encourage the authors to incorporate their additional experiments including comparisons to other baselines, running time of their approach, and reliability studies of the different parts of the pipeline.